# Amortized Fourier Neural Operators

**Zipeng Xiao**[1], **Siqi Kou**[1], **Zhongkai Hao**[2], **Bokai Lin**[1], **Zhijie Deng**[1]*
[1] Qing Yuan Research Institute, SEIEE, Shanghai Jiao Tong University
[2] Dept. of Comp. Sci. & Tech., Tsinghua University
{xiaozp_25, happy-karry}@sjtu.edu.cn, hzj21@mails.tsinghua.edu.cn,
{19821172068,zhijied}@sjtu.edu.cn

## Abstract

Fourier Neural Operators (FNOs) have shown promise for solving partial differential equations (PDEs). Typically, FNOs employ separate parameters for different frequency modes to specify tunable kernel integrals in Fourier space, which, yet, results in an undesirably large number of parameters when solving high-dimensional PDEs. A workaround is to abandon the frequency modes exceeding a predefined threshold, but this limits the FNOs' ability to represent high-frequency details and poses non-trivial challenges for hyper-parameter specification. To address these, we propose AMortized Fourier Neural Operator (AM-FNO), where an amortized neural parameterization of the kernel function is deployed to accommodate arbitrarily many frequency modes using a fixed number of parameters. We introduce two implementations of AM-FNO, based on the recently developed, appealing Kolmogorov–Arnold Network (KAN) and Multi-Layer Perceptrons (MLPs) equipped with orthogonal embedding functions respectively. We extensively evaluate our method on diverse datasets from various domains and observe up to 31% average improvement compared to competing neural operator baselines.

## 1 Introduction

Neural operators (NOs) have been extensively studied for their potential in accelerating the solving of partial differential equations (PDEs) in science and engineering fields [19, 20, 18, 24, 25, 2]. In contrast to approaches limited to specific discretizations or PDE instances [8, 36, 28, 29], NOs characterize the solving operator of a family of PDEs across different discretizations and hence enjoy higher efficiency and usability, e.g., for weather forecasting [26] and material analysis [5].

Fourier neural operator (FNO) [18] and its variants [17, 33, 31, 35] stand out as a significant subclass of NOs, which explore the convolution theorem and Fast Fourier Transform (FFT) to efficiently performs kernel integral, a central module for the learning operator of PDEs. Typically, FNO separately parameterizes the values of the Fourier-transformed kernel function for different frequency modes, and hinges on frequency truncation—abandons the parameters corresponding to frequencies exceeding some threshold—to reduce modeling costs, particularly for high-dimensional PDEs.

Frequency truncation can be problematic when solving PDE systems with intense high-frequency components. To address this, IFNO dynamically adjusts the threshold for frequency truncation during training, though it still experiences exponential parameter growth with increasing dimensionality [35]. This complexity can result in substantial memory consumption and hinder the development of large-scale pretrained models [9]. Conversely, AFNO [7] employs a shared MLP to transform the outcomes of FFT, but the uniform treatment of frequency modes may constrain expressiveness and lead to suboptimal performance.

---

*Corresponding author.

38th Conference on Neural Information Processing Systems (NeurIPS 2024).

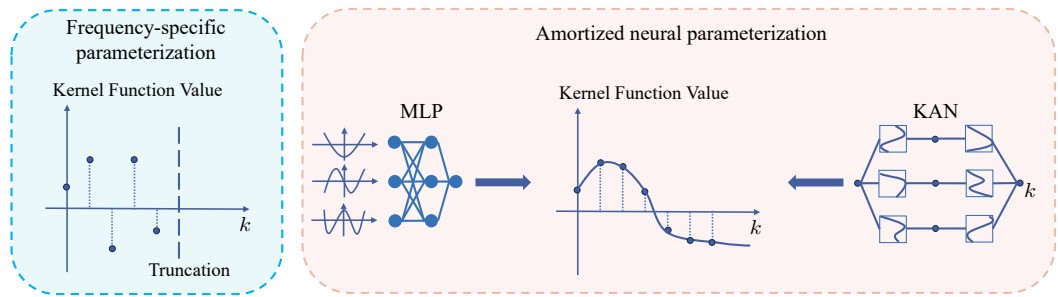

Figure 1: Comparison between FNO and AM-FNO: FNO assigns each value at the discretized frequencies of the Fourier-transformed kernel function as a learnable parameter, while AM-FNO utilizes neural network parameterization (MLP or KAN) to approximate the mapping between frequencies and function values. The frequencies are embedded using a set of orthogonal basis functions before being processed by the MLP.

We address this by developing *AMortized Fourier Neural Operator (AM-FNO)*, where we introduce extra neural networks (NNs) to specify the kernel integral operator to amortize the modeling cost. As illustrated in Figure 2, AM-FNO is simple and intuitive—learnable NN transformations are leveraged to directly define the Fourier-space kernel function to accommodate arbitrarily many frequency modes at the cost of a fixed number of parameters. An amortized parameterization also provides an inherent regularization mechanism for resisting overfitting to potential high-frequency noise.

The NN transformation in AM-FNO can be flexibly defined. One natural choice is the Kolmogorov–Arnold Network (KAN) due to its superior accuracy in function fitting [21]. Doing so, we, for the first time, reveal the potential of KAN for the operator learning of PDEs. Considering the widely criticized inefficiency issues of KAN in both time and memory consumption, we also investigate the regular Multi-Layer Perceptrons (MLPs) for amortized parameterization. We empirically identify the necessity of embedding the frequency modes with orthogonal basis functions before MLP transformation.

We experiment on challenging benchmarks governed by diverse typical PDEs, covering six standard PDE benchmarks [18, 17, 30]. AM-FNO (KAN) and AM-FNO (MLP) achieve an average 22% and 31% reduction in relative error on these benchmarks, respectively. We also analyze the error of different frequency modes, and the results show that our models outperform in all frequency ranges. Additionally, we perform zero-shot super-resolution on three benchmarks to assess the generalization ability of AM-FNO across discretizations. We observe that AM-FNOs outperform the baselines and achieve lower error even compared to FNO trained on the test resolution. The results reflect that AM-FNOs have the promising potential to substantially reduce both the data collection and training costs for solving high-resolution PDE problems.

## 2   Related Works

**Neural Operators.** Neural operators have attracted considerable interest for their capacity to map infinite-dimensional function spaces [24, 19, 1], thereby facilitating solutions across diverse discretizations without retraining. The pioneering work DeepONet [24] introduces a trunk and branch network architecture grounded in the universal approximation theorem for operators. Transformer-based neural operators represent a notable line of work in the domain. Galerkin Transformer [2] proposes self-attention operators that theoretically correspond to a learnable kernel integral operator and projection. OFormer [16] introduces an architecture with input and query encoders for querying arbitrary output locations. GNOT [10] proposes a heterogeneous normalized attention layer to encode different input information.

**FNO and its Variants.** Fourier neural operator (FNO) [18] represents a novel approach with exceptional efficiency and accuracy due to its ability to learn kernel integral operators in the Fourier domain. The theoretical proof establishes its capability to approximate arbitrary continuous operator [15]. Tailored for addressing multiphase flow problems, U-FNO [33] augments its representation in higher frequency information through U-Net paths. Geo-FNO [17] extends the applicability of FNO beyond uniform grids by employing a learnable mapping from irregular domains to uniform latent meshes.

F-FNO [31] reduces the model parameters and addresses performance degradation with increasing layers by factorizing the integral operator and enhancing the architecture. AFNO [7] transforms the function values after FFT with a shared MLP for each frequency, effectively reducing the parameter count. FNOs have made notable contributions across various challenging tasks [6, 25, 33, 26]. However, minimizing model complexity while effectively managing high-frequency information and ensuring generalization across different discretizations poses a persistent challenge in FNOs. Additionally, to the best of our knowledge, there has been no attempt to incorporate KANs with neural operators.

# 3  Preliminary

This section presents the foundations of operator learning and the integral operator used in FNO.

## 3.1  Operator Learning

Consider the input function space $\mathcal{A} = \mathcal{A}(D; \mathbb{R}^{d_a})$ and the target function space $\mathcal{U} = \mathcal{U}(D; \mathbb{R}^{d_u})$ defined on a bounded and open set $D \subset \mathbb{R}^d$. Operator learning seeks to learn a $\theta$-parameterized operator $\mathcal{G}_\theta$ to approximate the ground-truth mapping $\mathcal{G} : \mathcal{A} \to \mathcal{U}$ specified by a PDE. This learning process is based on a finite set of function observations $\{a_i, u_i\}_{i=1}^N$, where functions $a_i$ and $u_i$ are discretized on meshes $\{x_j \in D\}_{j=1}^M$. The optimization problem is:

$$\min_\theta \frac{1}{N} \sum_{i=1}^N \frac{\|\mathcal{G}_\theta(a_i) - u_i\|_2}{\|u_i\|_2}, \tag{1}$$

where the regular mean-squared error (MSE) is extended with a normalizer $\|u_i\|_2$ to handle scale variations across benchmarks, denoted as $l_2$ relative error. The relative error can also be substituted with other loss functions.

## 3.2  Fourier Integral Operator

A learnable kernel integral is a central module for defining mappings among functions. Specifically, we denote the hidden state of the input function in the $l$-th transformation stage as $h^{(l)}(y) : D \to \mathbb{R}^{d_h}$, where $d_h$ represents the dimensionality, assumed to be consistent across all stages. The kernel integral operator makes the following transformation:

$$(\mathcal{K}(h^{(l)}))(x) = \int_D \kappa(x, y) h^{(l)}(y) dy, \quad \forall x \in D \tag{2}$$

However, the integral is not computation-friendly within deep learning frameworks. To address this, FNO [18] assumes the kernel is shift-invariant, i.e., $\kappa(x, y) = \kappa(x - y)$, and leverages the convolution theorem to efficiently compute the integral in the Fourier domain:

$$(\mathcal{K}(h^{(l)}))(x) = \mathcal{F}^{-1}(R \cdot \mathcal{F}(h))(x), \quad \forall x \in D \tag{3}$$

where $\mathcal{F}$ and $\mathcal{F}^{-1}$ denote the FFT and its inverse (IFFT), and $R(k) : \mathcal{E} \to \mathbb{C}^{(d_h \times d_h)}$ represents the Fourier-transformed complex-valued kernel function with $k \in \mathcal{E}$ denoting a frequency mode. Typically, FNO individually parameterizes the values of $R(k)$ for a fixed range of frequency modes (denoting the number as $k_T$) to avoid high modeling costs. This, yet, limits the exploration of high-frequency details in the function and poses non-trivial challenges for hyper-parameter specification.

# 4  Method

This section elaborates on amortized FNO (AM-FNO), which amortizes an arbitrary number of frequency modes by sharing a fixed number of parameters.

## 4.1  Amortized Parameterization

Instead of parameterizing the kernel function point-by-point, we propose to build the mapping between frequency and the values of the Fourier-transformed kernel function using an NN. On one

hand, this mitigates the issue that the number of parameters increases significantly with that of frequency modes and dimensionality of PDEs. On the other hand, this approach avoids AFNO's uniform transformation, ensuring richer expressiveness in the Fourier domain.

Concretely, by the rule of Fourier transformation, there is:

$$R(k) = \int_D \kappa(x)e^{-2i\pi x \cdot k}dx. \tag{4}$$

Note that the output of $R$ is a matrix, so we can use $R_{p,q}, p, q \in \{1, 2, \ldots, d_h\}$ to denote the complex scalar-valued function yielding one element of the matrix output. Our AM-FNO directly uses NNs to define the real and imaginary parts of the function $R_{p,q}$ to maximize the modeling flexibility. Formally, there is

$$R_{p,q}(k) := \text{NN}_{re}(k) + \sqrt{-1}\,\text{NN}_{im}(k). \tag{5}$$

The detailed implementations of the two NNs have no essential difference, so we only discuss one of them in the following.

## 4.2 Kolmogorov-Arnold Networks

Kolmogorov-Arnold Networks (KANs) have been empirically shown to show promise in function approximation [21]. According to theoretical analysis, KANs are usually defined as:

$$\text{KAN}(\boldsymbol{x}) = \sum_{j=1}^{2H+1} \eta_j'(\sum_{t=1}^{H} \eta_{j,t}(x_t)) \tag{6}$$

where $\eta_j', \eta_{j,l} : \mathbb{R} \to \mathbb{R}$ denote the learnable basis functions. In practice, we can generalize the above definition and set a KAN layer as (with $\boldsymbol{x}$ as a $H$-dim variable):

$$\boldsymbol{x}' = \begin{pmatrix} \eta_{1,1}(\cdot) & \eta_{1,2}(\cdot) & \cdots & \eta_{1,H}(\cdot) \\ \eta_{2,1}(\cdot) & \eta_{2,2}(\cdot) & \cdots & \eta_{2,H}(\cdot) \\ \vdots & \vdots & \ddots & \vdots \\ \eta_{H',1}(\cdot) & \eta_{H',2}(\cdot) & \cdots & \eta_{H',H}(\cdot) \end{pmatrix} \boldsymbol{x}. \tag{7}$$

We can specify the function by learnable coefficients and multiple local B-spline basis functions [21]:

$$\eta(x) = w(r(x) + \text{spline}(x)), \quad \text{spline}(x) = \sum_g c_g B_g(x) \tag{8}$$

where $r(x)$ represents a basis function (typically sigmoid linear unit or SiLU [3]) and $w$ is a factor controlling the magnitude.

We utilize two two-layer KANs in each layer of AM-FNO to define the real and imaginary parts of $R_{p,q}$. The inputs to the network are all frequencies of $h^{(l)}$ after FFT. In fact, we can share weights among the KANs associated with $R_{p,q}$ with different $p$ and $q$. Empirical results (Table 4) indicate superior performance compared to the corresponding MLP implementation.

## 4.3 Multi-Layer Perceptrons

In practice, KANs require extensive training time. To address this, we propose an alternative parameterization aimed at improving the performance of Multi-Layer Perceptrons (MLPs). Despite its universal approximation ability, MLPs empirically suffer from compromising performance. The spectral bias of vanilla MLPs, i.e., their tendency to favor low-frequency functions, may limit their capacity to represent more complex functions. Motivated by the success of leveraging orthogonal basis functions for function approximation [12, 14, 27], we propose to augment the MLP with orthogonal embedding functions to construct AM-FNO.

Specifically, we can embed the frequency mode input using a set of orthogonal functions before the MLP transformation. The Fourier basis is a natural choice due to its capacity to capture high-frequency components, thereby enhancing the high-frequency representation of vanilla MLPs. However, our empirical results show that Chebyshev basis functions can perform better (see Table 4). Of note, we

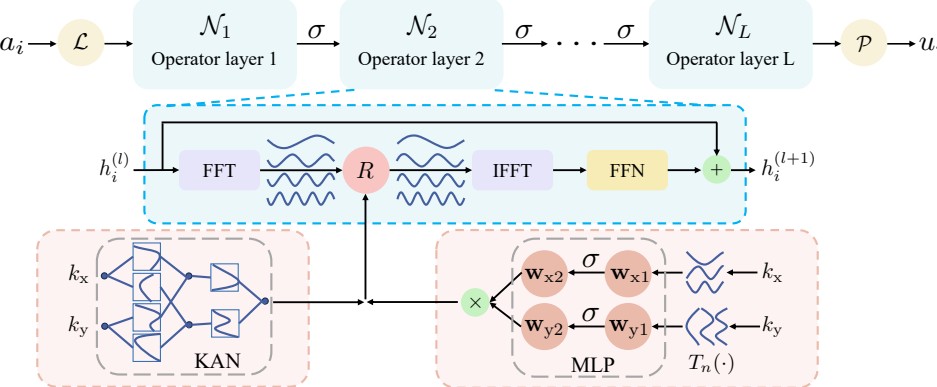

Figure 2: AM-FNO structure for 2D PDEs: The input function $a$ is mapped to a higher-dimensional space. Stacked operators and activation functions are applied for function propagation. Within the operator layers, a linear transformation $R$ is applied to $h^{(l)}$ after FFT, followed by a feed-forward network (FFN) after the Inverse Fast Fourier Transform (IFFT). The values of $R$ result from KAN or multiplying the MLP transformations of selected one-dimensional orthogonal basis functions (**w** denotes linear weights.). Finally, the function is projected to the solution dimension space.

use only the first $n_{\max}$ orthogonal basis functions in the family for the parameterization as we cannot employ infinite parameters.

**Factorization trick for high-dimensional PDEs.** For the Fourier-transformed kernel, the dimensionality of the frequency modes matches that of the PDE being solved. To approximate functions with $d$-dimensional inputs, a common approach is to construct $d$-dimensional orthogonal functions based on one-dimensional basis functions. Specifically, for $n_{\max}$ one-dimensional basis functions in each of the $d$ dimensions, the complete space contains $n_{\max}^d$ high-dimensional basis functions, resulting in an exponential complexity for modeling.

Motivated by the dimension factorization in the integral operator [31], we separate the input dimensions and construct the final kernel by the products of the kernel approximated in each dimension. Such a process is illustrated in Figure 2. In this way, the total parameter count scales linearly w.r.t. the dimension of the PDEs.

## 4.4 AM-FNO Architecture

In FNO, a stacked structure is employed to approximate the entire mapping $\mathcal{G}$, as illustrated below:

$$\mathcal{G} = \mathcal{P} \circ \mathcal{Q}^{(L)} \circ \cdots \circ \mathcal{Q}^{(1)} \circ \mathcal{L} \tag{9}$$

where $\mathcal{L}$ maps the input function $a$ to the hidden state $h^{(0)}$, and $\mathcal{P}$ maps the hidden state $h^{(L)}$ to the output function $u$, both in a point-wise manner. $\mathcal{Q} : h^{(l)} \to h^{(l+1)}$ is the operator layer responsible for the iterative update:

$$\mathcal{Q}(h^{(l)}) = \sigma(W h^{(l)} + \mathcal{K}(h^{(l)}) + b) \tag{10}$$

where $\mathcal{K}$ represents the kernel integral operator in Equation (3), and $b$ represents the bias. FNO employs a pointwise linear mapping with $W$ to enable the propagation of high-frequency information, but this is empirically limited, validated by the results in Table 8.

Given the non-truncated Fourier transform of the kernel function $R$ in AM-FNO, we employ the operator layer architecture in [31], which replaces the linear map $W$ with residual connection [11]:

$$\mathcal{N}(h^l) = h^{(l)} + (W_2\sigma(W_1\mathcal{K}(h^{(l)}) + b_1) + b_2). \tag{11}$$

As shown in Figure 2, we also incorporate activation functions between the operator layers for enhanced flexibility. The resultant model structure is:

$$\mathcal{G} := \mathcal{P} \circ \mathcal{N}^{(L)} \circ \sigma \circ \mathcal{N}^{(L-1)} \circ \sigma \circ \cdots \circ \mathcal{N}^{(1)} \circ \mathcal{L}. \tag{12}$$

Table 1: Overview of benchmarks including their spatial dimensions $d$, spatial resolution $M$, temporal resolution $N_t$, and training data $N_{\text{train}}$ and test data $N_{\text{test}}$.

| Benchmark | PDE | Geometry | $d$ | $M$ | $N_t$ | $N_{\text{train}}$ | $N_{\text{test}}$ |
|---|---|---|---|---|---|---|---|
| Darcy | Darcy flow | Regular grid | 2 | $85 \times 85$ | - | 1000 | 200 |
| NS-2D | Navier-Stokes | Regular grid | 2 | $64 \times 64$ | 20 | 1000 | 200 |
| Pipe | Navier-Stokes | Structured mesh | 2 | $129 \times 129$ | - | 1000 | 200 |
| Airfoil | Euler equation | Structured mesh | 2 | $221 \times 51$ | - | 1000 | 200 |
| Elasticity | Elastic Wave | Point cloud | 2 | 972 | - | 1000 | 200 |
| CFD-1D | Compressible Navier-Stokes | Regular grid | 1 | 128 | 21 | 1800 | 200 |
| CFD-2D | Compressible Navier-Stokes | Regular grid | 2 | $64 \times 64$ | 21 | 1800 | 200 |

## 5 Experiment

In this section, we validate the effectiveness of our proposed method by conducting extensive experiments on challenging benchmarks governed by typical solid and fluid PDEs.

### 5.1 Experimental Setup

**Benchmarks.** We evaluate the performance of AM-FNO on six well-established benchmarks. These benchmarks include Burger, Darcy, and NS-2D, which are presented in regular grids with varying dimensions [18]. We extend our experiments to assess the method's performance in different geometries, including Pipe, Airfoil, and Elasticity benchmarks [17] Additionally, we incorporate the compressible fluid dynamics (CFD) 1D and 2D benchmarks [30], which involves more high-frequency information. The benchmarks are summarized in Table 1.

**Baselines.** We conduct a comparative evaluation of our neural operator against seven baseline methods. These baselines include well-recognized approaches such as FNO [18] and its variants Geo-FNO [17], U-FNO [33], F-FNO [31] and AFNO [7]. Additionally, we consider other models, including OFormer [16] and LSM [34]. Notably, LSM represents the latest state-of-the-art (SOTA) neural operator among the baselines.

**Implementation details.** We train all models for 500 epochs using the AdamW optimizer [23] with a cosine annealing scheduler [22]. The initial learning rate is $10^{-3}$, and the weight decay is set to $10^{-4}$. Our models consist of 4 layers with a width of 32 and process all the frequency modes of training data. The Gaussian Error Linear Unit (GELU) is used as the activation function [13]. For AM-FNO (KAN), the number of spline grids is selected from $\{24, 32, 48\}$, while for AM-FNO (MLP), the number of basis functions is set to 32 or 48. AM-FNO (MLP) utilizes Chebyshev basis functions as the orthogonal basis functions, as elaborated in the appendix. The batch size is selected from $\{4, 8, 16, 32\}$, and the experiments are conducted on a single 4090 GPU. The evaluation metric and training loss are based on the $l_2$ relative error in Equation (1), unless otherwise specified. We employ the transformation method from geo-FNO [17] to map between irregular input domains and uniform meshes for the Elasticity benchmark on point clouds. More details about the baselines can be found in Appendix A.

### 5.2 Main Results

The main results are shown in Table 2. Our models consistently achieve state-of-the-art (SOTA) performance on all six benchmarks with various PDEs, geometries, and dimensions. AM-FNOs exhibit a significant average performance improvement of 22% from KAN implementation and 31% from MLP implementation compared to the top-performing baseline. We provide a comparison of GPU memory and training time in Appendix D, showing that although AM-FNO retains all frequency modes, it achieves comparable memory usage and training time to other FNOs.

AM-FNOs significantly reduce prediction error on Darcy and NS-2D benchmarks, standard benchmarks with strong low-frequency components. Specifically, AM-FNO (KAN) reduces the error by 39% (2.73e-3) and 11% (1.40e-2), while AM-FNO (MLP) reduces the error by 40% (2.80e-3) and 30% (3.69e-2). This improvement can be attributed to AM-FNO's kernel parameterization, which effectively captures the low-frequency information. Additionally, AM-FNO demonstrates robustness across irregular geometries, with reductions of 5% (3.30e-4), 32% (1.66e-3) and 7%(1.5e-3) from AM-FNO (KAN), and 12% (7.50e-4), 34% (1.76e-3) and 10% (2.20e-3) from AM-FNO (MLP) in

Table 2: Comparison of the primary findings across six benchmark tests with six baseline methods. Lower scores signify superior performance, with the best outcome highlighted in bold and the second-best outcome underlined. The presence of a "- " indicates that the corresponding baseline is incapable of addressing the benchmark.

| Model | Darcy | NS-2D | Pipe | Airfoil | Elasticity | CFD-1D | CFD-2D |
|---|---|---|---|---|---|---|---|
| FNO | 1.08e-2 | 1.56e-1 | - | - | - | 2.93e-2 | 5.36e-3 |
| AFNO | 3.17e-2 | 2.17e-1 | 1.72e-2 | 9.88e-3 | 4.57e-2 | - | 6.72e-2 |
| Geo-FNO | 1.08e-2 | 1.56e-1 | 6.70e-3 | 1.38e-2 | 2.29e-2 | 2.93e-2 | 5.36e-3 |
| OFormer | 1.24e-2 | 1.71e-1 | 9.59e-3 | 1.83e-2 | - | - | - |
| U-FNO | 1.24e-2 | 1.22e-1 | 5.76e-3 | 1.05e-2 | 2.26e-2 | 2.44e-2 | 4.52e-3 |
| F-FNO | 9.92e-3 | 1.74e-1 | 5.99e-3 | 1.00e-2 | 3.16e-2 | 2.54e-2 | 7.86e-3 |
| LSM | 7.01e-3 | 1.64e-1 | 5.20e-3 | 6.39e-3 | 2.25e-2 | - | 7.62e-2 |
| Ours (KAN) | 4.28e-3 | 1.08e-1 | 3.54e-3 | 6.06e-3 | 2.10e-2 | 1.83e-2 | 2.70e-3 |
| Ours (MLP) | **4.21e-3** | **8.51e-2** | **3.44e-3** | **5.64e-3** | **2.03e-2** | **1.47e-2** | **2.16e-3** |

prediction error on Airfoil, Pipe and Elasticity benchmarks. For CFD-1D and CFD-2D benchmarks characterized by stronger high-frequency components, AM-FNO (KAN) achieves improvements of 25% (6.10e-3) and 40% (1.83e-3), while AM-FNO (MLP) achieves improvements of 40% (9.70e-3) and 52% (2.36e-3). The promotion highlights the effectiveness of our method of handling high-frequency components. We also find that AM-FNO (MLP) outperforms AM-FNO (KAN) across all benchmarks, likely due to the enhanced expressiveness of the orthogonal embedding.

## 5.3 Frequency-Based Error Analysis

**CFD-1D.** To assess the performance across various frequency modes, we calculate the error of different frequency modes after FFT on CFD-1D benchmark and incorporate FNO without truncation (FNO$^{+}$) for comparison. The results are visualized in Figure 3.

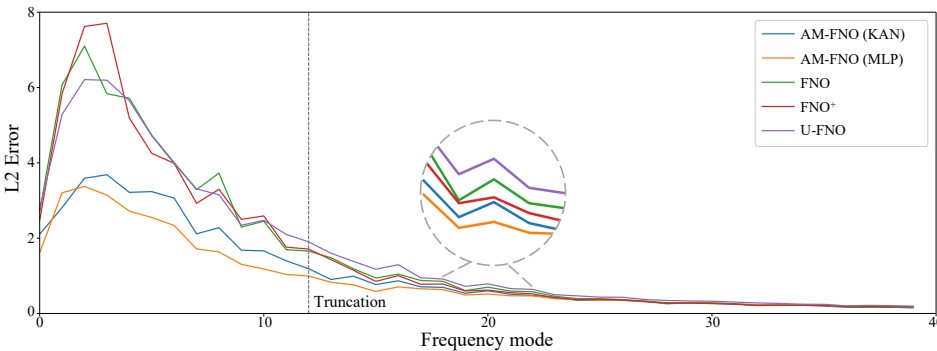

Figure 3: Comparison of L2 norm error on different frequency modes on CFD-1D benchmark.

As shown, the errors primarily stem from the first few frequency modes and decrease as the frequency increases. The baselines exhibit similar errors in the truncated frequency range, whereas our models demonstrate significantly lower errors. Our models maintain an advantage over the baselines in the initial modes within the truncated frequency range. As the frequency increases, the strength of the high-frequency components diminishes, and all the errors become negligible.

**CFD-2D.** We further evaluate the performance across various frequency modes with the metrics outlined in [30] (detailed in Appendix B) and present the results on CFD-2D benchmark in Table 3.

The results showcase that FNO$^{+}$ demonstrates lower train error but higher test error than FNO and U-FNO. Notably, FNO$^{+}$ exhibits similar prediction errors to FNO in the high-frequency range, suggesting potential overfitting to the training data. We propose that the substantial complexity parameterization of FNO$^{+}$ may render it sensitive to high-frequency details, limiting the effectiveness of the additional parameters in handling such components. The fL2 error of U-FNO across all frequency ranges is lower than FNO, which can be attributed to the enhanced expressiveness enabled by the additional U-Net architecture. AM-FNO (MLP) achieves the lowest training error and fL2 error

Table 3: Comparison of the error in different frequency regions on CFD-2D benchmarks. Each complex-valued parameter is considered as 2 in the parameter count (Param). Train error (Train Err.) and test error (Test Err.) are evaluated using the $l_2$ relative error at each time step. fL2 signifies the $l_2$ relative error in Fourier space (fRMSE) pertaining to the low, middle, and high-frequency regions. FNO$^+$ refers to FNO without frequency truncation.

| Model | Param (M) | CFD-2D | | | | |
|---|---|---|---|---|---|---|
| | | Train Err. | Test Err. | fL2 low | fL2 mid | fL2 high |
| FNO | 2.37 | 2.57e-3 | 5.36e-3 | 1.51e-3 | 7.66e-1 | 1.49e-1 |
| FNO$^+$ | 18.39 | 1.63e-3 | 5.81e-3 | 1.82e-3 | 7.31e-1 | 1.42e-1 |
| U-FNO | 2.66 | 2.00e-3 | 4.52e-3 | 1.26e-3 | 6.55e-1 | 1.28e-1 |
| Ours (KAN) | 2.21 | 1.79e-3 | 2.70e-3 | 7.78e-4 | 4.19e-1 | 7.65e-2 |
| Ours (MLP) | 2.29 | **1.34e-3** | **2.16e-3** | **6.22e-4** | **3.55e-1** | **5.76e-2** |

Table 4: Comparison of the $l_2$ relative error for different components of AM-FNO (MLP) on Darcy, Airfoil, and Pipe benchmarks. Chebyshev basis functions are substituted with triangular basis functions (TBF) and non-orthogonal polynomial basis functions (PBF). A version of the model without orthogonal embedding (Non) is included for comparison. The training time and memory requirements are derived from the Airfoil benchmark.

| Designs | Param (M) | Mem (MB) | Time (s/epoch) | Darcy | Airfoil | Pipe |
|---|---|---|---|---|---|---|
| TBF | 1.14 | 1890 | 2.61 | **4.13e-3** | 7.60e-3 | 3.80e-3 |
| PBF | 1.13 | 1890 | 2.55 | 1.77e-2 | 1.30e-2 | 1.03e-2 |
| Non | 1.10 | 1826 | 2.70 | 1.29e-2 | 7.21e-3 | 7.84e-3 |
| Ours (MLP) | 1.14 | 1890 | 2.52 | 4.21e-3 | **5.64e-3** | **3.44e-3** |
| Ours (KAN) | 1.56 | 2230 | 4.70 | 4.28e-3 | 6.06e-3 | 3.54e-3 |

across all frequency ranges, while AM-FNO (KAN) achieves the second lowest fL2 error. Specifically, AM-FNO (MLP) and AM-FNO (KAN) achieve 55%(7.04e-2) and 50%(5.15e-2) reduction in the high-frequency range. This outcome can be attributed to our amortized parameterization, which significantly reduces model complexity while maintaining adequate expressiveness to approximate the Fourier-transformed kernel function.

## 5.4 Ablation Experiments

We conduct a detailed ablation study to assess the effectiveness of different components and hyperparameters of our models.

**Necessity of Orthogonal Embedding.** We study the impact of orthogonal embedding on Darcy, Airfoil, and Pipe benchmarks. Table 4 presents the findings. Although AM-FNO (KAN) outperforms the MLP version without embedding (Non), its efficiency is significantly lower than that of versions utilizing MLPs. Removal of the orthogonal embedding leads to a notable performance decline across all three benchmarks. Meanwhile, replacing the orthogonal functions with non-orthogonal polynomial basis functions results in the highest prediction error among the baseline models. These outcomes showcase the efficacy and indispensability of the orthogonal embedding. To validate the robustness of the embedding, we use triangular basis functions and observe comparable errors on the Darcy and Pipe benchmarks, but higher errors on the Airfoil benchmark. We attribute this difference to the accuracy of function approximation achieved by Chebyshev basis functions.

**Influence of Some Hyperparameters** We conduct experiments to assess how prediction error varies with different numbers of basis functions, hidden sizes of KANs, and grid sizes of splines (linearly scaled with local spline count) in KANs. Figure 4 shows the results. The left figure illustrates that the error decreases with an increasing number of basis functions in orthogonal embedding on both benchmarks. This reduction is particularly evident initially on the Airfoil benchmark, followed by diminishing returns. In this case, employing 24 basis functions achieves a favorable balance between efficiency and accuracy. This performance enhancement can be attributed to the increased

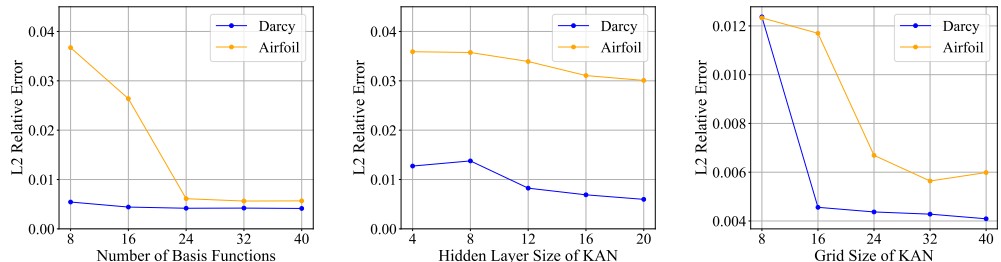

Figure 4: $l_2$ relative error varies w.r.t. the number of basis functions (Left), hidden layer size of KANs (middle), and grid size of KANs (right) on Darcy and Airfoil benchmarks.

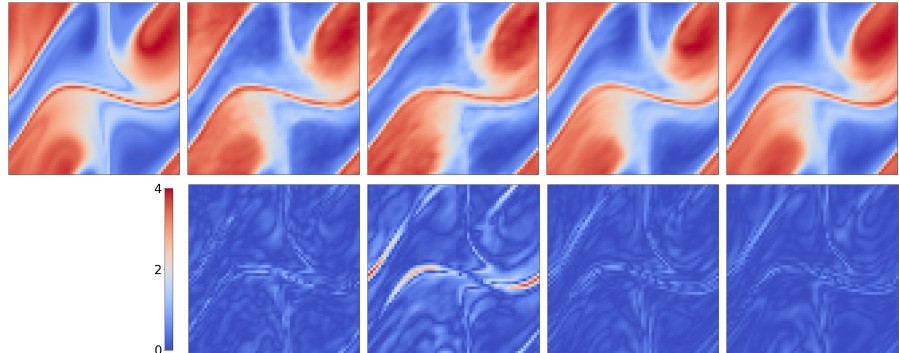

Figure 5: Comparison of zero-shot super-resolution absolute errors on NS-2D benchmark. The top row displays the ground truth, FNO, U-FNO, AM-FNO (KAN), and AM-FNO (MLP) predictions from left to right. The bottom row illustrates their errors.

expressiveness from the additional orthogonal basis functions. In the middle figure, while the error decreases with increasing hidden size, we observe a decline in performance compared to KANs with larger grid sizes but smaller hidden sizes. This may suggest that the expressive power of KANs is primarily derived from the number of local spline functions (grid size). The right figure displays a trend similar to the left one: a noticeable decrease initially, followed by a less pronounced reduction later. In this study, we suggest increasing the grid size to enhance performance rather than focusing on adjusting the hidden layer size. We also present the performance of AM-FNOs, retaining the same frequency modes as other FNOs, in Table 9. AM-FNOs consistently outperform baseline models, underscoring the advantages of our amortized parameterization over the standard FNO approach.

## 5.5 Zero-Shot Super-Resolution

A notable characteristic of neural operators is their ability to generalize across various discretizations. We conduct experiments training on lower resolution data and evaluate on higher resolution data on NS-2D benchmark. We visualize the results in Figure 5 and provide the numerical results in Table 5.

U-FNO exhibits a significant performance degradation when evaluated on higher resolution. This decline can be attributed to the convolutional operation in the U-Net architecture, which possesses a fixed receptive field and cannot effectively generalize across different discretizations. In contrast, FNO and AM-FNOs demonstrate the

| M | FNO | U-FNO | Ours (KAN) | Ours (MLP) |
|---|---|---|---|---|
| $32 \times 32$ | 1.32e-1 | 1.39e-1 | 9.84e-2 | 8.79e-2 |
| $64 \times 64$ | 1.28e-1 | 2.18e-1 | 9.96e-2 | 8.73e-2 |

Table 5: Comparison of $l_2$ relative error across different resolutions on NS-2D benchmark, with all models trained with $32 \times 32$ resolution.

ability to generalize across different discretizations. Notably, AM-FNOs achieve superior performance at $64 \times 64$ resolution compared to baselines trained with the same resolution (see Table 2), which underscores the data efficiency of AM-FNOs.

# 6 Conclusion

This paper proposes AM-FNOs to improve Fourier neural operator (FNO)'s efficiency in addressing PDEs without frequency truncation. Our approach utilizes Kolmogorov–Arnold Networks (KANs) and Multi-Layer Perceptrons (MLPs) with orthogonal embedding functions to mitigate exponential complexity and overfitting to high-frequency noise. Comprehensive experiments across various datasets demonstrate the effectiveness of AM-FNOs compared to baseline approaches.

**Limitations.** This work attempts to enhance FNO's handling of high-frequency information but has the following limitations. The benchmarks used are idealized physical systems, excluding real-world complex problems. Meanwhile, although AM-FNOs reduce the parameter count, the extremely high-dimensional PDEs still pose challenges for FNOs due to the complexity of FFT.

## Acknowledgements

This work was supported by NSF of China (No. 62306176), Natural Science Foundation of Shanghai (No. 23ZR1428700), CCF-Zhipu Large Model Innovation Fund (No. CCF-Zhipu202412), and CCF-Baichuan-Ebtech Foundation Model Fund.

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

## A  Hyperparameters and Details for Models

**AM-FNOs.** For AM-FNO (MLP), Chebyshev polynomials are chosen due to their favorable theoretical accuracy [32, 4]. Formally, they are defined as:

$$\text{For } k \in [-1, 1], \quad T_0(k) = 1, \quad T_1(k) = k,$$
$$T_{n+1}(k) = 2kT_n(k) - T_{n-1}(k) \quad n \geq 1. \tag{13}$$

The polynomial demonstrates an increase in the degree of its highest power term concerning $k$ as $n$ increases. It can be alternatively formulated as:

$$T_n(k) = \cos(n\arccos(k)), \quad \text{for } k \in [-1, 1]. \tag{14}$$

For AM-FNO (KAN), we fix the grid size during training to ensure a consistent parameter count. The order of the spline is fixed as 3.

**FNO and its Variants.** We employ 4 layers with modes set to 12 and widths set to 32 for FNO and its variations (Geo-FNO, U-FNO, F-FNO). U-FNO incorporates the U-Net path in the last two layers. For AFNO, we set the width to 512 and use 4 layers to maintain a comparable parameter count. Notably, Geo-FNO reverts to the vanilla FNO when applied to benchmarks with regular grids, resulting in equivalent performance for Darcy and NS-2D benchmarks.

**LSM.** The model is employed with 8 basis operators and 4 latent tokens. The width of the first scale is set to 32, with a downsampling ratio of 0.5.

**OFormer** The depth of the encoder is fixed at 6, while the hidden dimension is set to 96.

## B  Metrics

Following [30], we use fL2 error to quantify errors in different frequency ranges. It is computed as:

$$\frac{\|\mathcal{F}(u_{pred}) - \mathcal{F}(u_{true})\|_2}{\|\mathcal{F}(u_{true})\|_2} \tag{15}$$

where $\mathcal{F}$ denotes the FFT and the frequency range is restricted to $k_{\min} \leq k \leq k_{\max}$. For fL2 low, $k_{\min} = 0$ and $k_{\max} = 4$; for fL2 mid, $k_{\min} = 5$ and $k_{\max} = 12$; for fL2 high, $k_{\min} = 12$ and $k_{\max} = \infty$.

## C Comparison of GPU Memory, Training Time, and Parameter Counts.

Table 6: Comparison of GPU memory, training time per epoch, and parameter counts on Darcy benchmark.

| Model | AM-FNO(MLP) | AM-FNO(KAN) | FNO | U-FNO | OFormer | LSM | F-FNO | AFNO |
|---|---|---|---|---|---|---|---|---|
| Train Time (s) | 1.11 | 2.1 | 0.9 | 1.6 | 16.5 | 2.1 | 1.1 | 27.3 |
| Memory (M) | 1850 | 2066 | 1212 | 1444 | 16090 | 1894 | 1126 | 11288 |
| Params. (M) | 1.1 | 1.5 | 2.4 | 2.6 | 1.3 | 4.8 | 0.2 | 2.6 |

## D Repeated results on NS-2D and CFD-2D benchmarks.

Table 7: Comparison of the $l_2$ relative error on NS-2D and CFD-2D benchmark.

| Benchmark | AM-FNO(MLP) | AM-FNO(KAN) |
|---|---|---|
| NS-2D | 8.53e-2 $\pm$ 7.48e-4 | 1.04e-1 $\pm$ 3.27e-3 |
| CFD-2D | 2.21e-3 $\pm$ 4.13e-5 | 2.75e-3 $\pm$ 8.96e-5 |

## E Abltation Study

We investigate the impact of the dimensional factorization trick and the model architecture. The results are shown in Table 8.

Table 8: Comparison of the $l_2$ relative error for different components of AM-FNO (MLP) on Darcy, Airfoil, and Pipe benchmarks. The version with the vanilla FNO architecture (Vanilla) and the version without dimensional factorization (No-DF) are included. The training time and memory requirements are derived from the Airfoil benchmark.

| Designs | Param (M) | Mem (MB) | Time (s/epoch) | Darcy | Airfoil | Pipe |
|---|---|---|---|---|---|---|
| Vanilla | 1.11 | 1274 | 2.28 | 5.25e-3 | 8.03e-3 | 3.89e-3 |
| No-DF | 1.10 | 1938 | 2.61 | 5.24e-3 | 5.89e-3 | 3.83e-3 |
| Ours (MLP) | 1.14 | 1890 | 2.52 | 4.21e-3 | **5.64e-3** | **3.44e-3** |
| Ours (KAN) | 1.56 | 2230 | 4.70 | 4.28e-3 | 6.06e-3 | 3.54e-3 |

The results indicate that the version without dimensional factorization performs comparably. Dimensional factorization can be unnecessary for handling low-dimensional PDEs. Meanwhile, the employed architecture achieves an average error reduction of $20\%$.

We also report the performance of AM-FNOs retaining the same number of frequency modes as other FNOs (12) on Darcy, Airfoil, and Elasticity benchmarks. The results, shown in Table 9, indicate that AM-FNOs consistently outperform the baselines, highlighting the advantages of our amortized parameterization.

## F Discussion about KAN and MLP.

Our findings in Section 5 demonstrate that implementing MLPs with orthogonal embeddings results in superior accuracy and efficiency. We attribute this success to the efficacy of the embedding technique employed. Meanwhile, the compatibility of the KAN architecture with widely used optimizers, such as AdamW, remains questionable. However, AM-FNO (KAN) offers several benefits over AM-FNO (MLP). First, as illustrated in Figure 4, the expressiveness of KANs primarily stems from the grid size. To uphold a constant parameter count, the grid size remains unchanged. However, the architecture is inherently extensible during training, which could potentially enhance accuracy. Second, KANs provide a level of interpretability, as discussed in [21], which holds significant value within this domain. Third, there is no need to select orthogonal basis functions for AM-FNO (KAN).

Table 9: Comparison of the $l_2$ relative error on Darcy, Airfoil, and Elasticity benchmarks.

| Benchmark | AM-FNO(MLP) | AM-FNO(KAN) | Geo-FNO | U-FNO | OFormer | LSM | F-FNO |
|---|---|---|---|---|---|---|---|
| Darcy | 4.72e-3 | 4.78e-3 | 1.08e-2 | 1.24e-2 | 1.24e-2 | 7.01e-3 | 9.92e-3 |
| Airfoil | 6.26e-3 | 6.62e-3 | 1.38e-2 | 1.05e-2 | 1.83e-2 | 6.59e-3 | 1.00e-2 |
| Elasticity | 2.10e-2 | 2.10e-2 | 2.29e-2 | 2.26e-2 | - | 2.25e-2 | 3.16e-2 |

## G    Impact Statements

This study introduces a neural operator specifically designed for the effective solution of partial differential equations (PDEs), potentially contributing to advancements in scientific and engineering domains. Positioned as foundational research in machine learning, the immediate identification of negative consequences is not evident, and the current risk of misuse remains low.

