# OpenReview forum: "Amortized Fourier Neural Operators"
_NeurIPS.cc/2024/Conference — NeurIPS 2024 poster_

### Official Review · Reviewer_kDgT · 2024-06-28

**Soundness:** 3
**Presentation:** 2
**Contribution:** 3
**Rating:** 5
**Confidence:** 4

**Summary:**

This paper tries to alleviate one of the issues in a so-called "Fourier Neural Operator (FNO)" which is a machine learning model to estimate a solution of partial differential equations (PDEs) based on the concept of "Operator learning". FNO takes into account the embedding of input field information in Fourier space, which partially contributes to its very weak dependence on the resolution. However, to reduce numerical cost, FNO abandons the frequency modes exceeding a predefined threshold, which limits its expressiveness power in the high-frequency region. To alleviate this issue, this paper proposed "Amortized Fourier Neural Operator (AM-FNO)", which utilizes a neural kernel function to accommodate arbitrarily many frequency modes with a fixed number of parameters, resulting in more expressive power with moderate model size.

**Strengths:**

1. The proposed AM-FNO allows to accommodate arbitrarily many frequency modes with a fixed number of parameters, resulting in more expressive power with moderate model size. Numerical experiments indicate that it improves not only the high-frequency regime but most regimes (mainly the low-frequency regime, surprisingly (Fig. 3))

2. The experiments utilize various datasets covering diverse PDEs with 1D and 2D spatial coordinate systems.

3. AM-FNO keeps a near-resolution independent nature of FNO even though considering high-frequency information.

**Weaknesses:**

The following major weaknesses are the factors forcing me to give a relatively lower score on this paper:

Major weakness:

1. Each experiment in the paper is conducted only once, which prohibits readers (and reviewers) from distinguishing whether the obtained improved performance is either due to the true-effectiveness of the proposed approach or just a lucky statistical fluctuation of optimizer and initial model weight. In particular, the training sample number is 1000, which is relatively small and can result in large statistical fluctuation. The justification in "Checklist" says that this is due to "computational cost", which does not validate anything but sounds like the authors lazily submit this version of the paper just for the deadline (I hope not). Following the ML convention, averaging over three to five times experiments with the standard deviation information to the result table are necessary before acceptance. I feel this crucial defect unnecessarily decreases the worth of this paper, though the proposed method seems important and very interesting.
At least, NS-2D and CFD-2D results in Table 2 should be given with standard deviation values because of their solution complexity, which can cause strong statistical fluctuations.

2. No information on the validation dataset is provided. Explain how to record the best score. If not using the validation dataset, provide a clear explanation of not overfitting to the test dataset.

Minor:
1. "related works" only introduces the work relating to the neural operators, which covers a part of ML for PDE. The authors should introduce the other ML models for PDE and explain why picked up the neural operator and FNO in this paper (maybe in Appendix).

2. There are several undefined symbols and terms, in particular, in Section 3, such as d_a, d_u, d (seemingly already observed in FNO paper..), and FNN in the caption of Figure 2.

3. The description of "Factorization trick for high-dimensional PDEs" seems too short. I encourage the authors to provide a more comprehensive explanation with mathematical descriptions either in the main-body or appendix.

4. No discussion on the training/inference time and memory consumption required for AM-FNO in comparison to FNO.

**Questions:**

1.  In sec 3.1 above of Eq. (1), what does it mean: "functions a_i and u_i"? Do the authors want to indicate that each numerical cell at i owns independent functions?

2. In Eq. (5), the authors use NN(k) for both real and imaginary parts. Do the authors want to indicate the real and imaginary parts share the same weight? (it seems not). It would be better to describe them as NN(k; \theta_real) and NN(k; \theta_imaginary).

3. Below Eq. (8), lines 129-130, the description seems to contradict the result in Table 4 (KAN is worse than MLP). Do the authors indicate the case "Non"? Seemingly, the description is insufficient.

4. Why have the authors introduced the well-known "Stone-Weierstrass theorem" [1,2] (or Weierstrass approximation theorem) as Theorem 4.1 without citing it? Does the authors' version include non-trivial improvement? In addition, the function f should not be an "arbitrary function" but a "continuous function".

          [1] Stone, Marshall Harvey. "Applications of the theory of Boolean rings to general topology." Transactions of the American Mathematical Society 41.3 (1937): 375-481.
          [2] Stone, Marshall H. "The generalized Weierstrass approximation theorem." Mathematics Magazine 21.5 (1948): 237-254.

5. (suggestion) AM-FNO(KAN) can be moved into the appendix to increase the space to explain other important information because AM-FNO(KAN) is consistently worse than AM-FNO(MLP).

**Limitations:**

The authors addressed the limitation in the manuscript.

---

> ### Author Rebuttal · Authors · 2024-08-07
>
> We thank Reviewer kDgT for the detailed feedback. Below, we respond to the questions.
>
> W1: Experimental repetition.
>
> The results for neural operators are relatively stable. Consequently, many results for widely used baselines are taken directly from the original papers, which also do not include repeated experiments [1,2]. Reproducing experiments for every baseline and benchmark multiple times is computationally intensive. To address this concern, we have conducted additional experiments with three repetitions for our method on NS-2D and CFD-2D benchmarks as shown in the table below. The standard deviations are minimal compared to the average values. We will include these results, along with standard deviation values, in the revised manuscript to better assess the statistical stability of our approach. We appreciate your feedback and will incorporate these improvements into the revised version.
>
> |Benchmark|AM-FNO(MLP)|AM-FNO(KAN)|
> |------|--------|---------|
> |NS-2D|8.53e-2 ± 7.48e-4|1.04e-1± 3.27e-3|
> |CFD-2D|2.21e-3 ± 4.13e-5|2.75e-3 ± 8.96e-5|
>
> W2: Validation dataset.
>
> The primary objective of our experiments is to evaluate the effectiveness of our models across various benchmarks and to provide a comparative analysis with existing baselines. To ensure fairness and consistency, we use a fixed set of training settings in line with other baseline studies [1,2,3]. We do not perform hyperparameter tuning specifically for individual benchmarks in Table 2, as detailed in Appendix B. This approach aligns with standard practices in operator learning and helps mitigate the risk of overfitting by preventing excessive optimization for any single benchmark.
>
> W3: Related works.
>
> We have briefly discussed our choice of neural operators over other neural network-based methods in the introduction. Due to space constraints, a more detailed comparison with other ML models for PDEs will be included in the revised version, potentially in the appendix.
>
> W4: Undefined symbols.
>
> Thank you for pointing this out. In Section 3, $d_a$ and $d_u$ represent the dimensions of different functions, and FFN refers to the feed-forward neural network. We will provide detailed definitions for these terms in the revised version.
>
> W5: More description of the factorization trick.
>
> The "Factorization trick for high-dimensional PDEs" was briefly introduced as it serves primarily as a supplementary technique rather than a core component of our method. We will provide a more detailed mathematical description and explanation in the appendix.
>
> W6: Training/inference time and memory consumption.
>
> We present a comparison of training/inference times and memory consumption for AM-FNO and FNO on the 2D Darcy benchmark with a resolution of $421 \times 421$ in the table below. The results indicate that AM-FNO (MLP) exhibits both reduced memory usage and shorter training times compared to FNO, which is attributable to its lower complexity. Although AM-FNO (KAN) demonstrates increased training time due to its architectural design, it still benefits from lower memory consumption. We conjecture that this advantage is particularly evident when solving PDEs with high resolution and high dimensions. During inference, AM-FNOs (with kernels generated by MLP or KAN being precomputed) exhibit a similar speed to FNO, as both methods rely on similar kernel calculations. We will provide a more detailed discussion of these aspects in the revised version of the paper.
>
> |Model|Memory|Train Time|Inf Time|
> |-----|------|------|------|
> |AM-FNO(MLP)|**9.7G**|**43.1s**|2.4s|
> |AM-FNO(KAN)|13.5G|83.8s|2.2s|
> |FNO|14.9G|45.6s|2.2s|
>
> Q1: $a_i$ refers to different input functions indexed by $i$, while $u_i$ denotes the corresponding output functions.
>
> Q2: Thanks for your suggestion. We will describe them differently in the revised version.
>
> Q3: We apologize for the confusion. The caption states, "A version of the model without orthogonal embedding (Non) is included for comparison," which refers to the model that directly approximates the kernel with MLP, as opposed to the KAN approach.
>
> Q4: Citation of "Stone-Weierstrass theorem".
>
> The Stone-Weierstrass theorem specifically addresses polynomial approximation of continuous functions, while our theorem generalizes approximation using orthogonal function bases [4], not limited to orthogonal polynomials. We will add the appropriate reference and clarify this distinction in the revised manuscript.
>
> Q5: Moving the KAN part.
>
> Thank you for your suggestion. While AM-FNO(KAN) does not outperform AM-FNO(MLP), it offers significant advantages, such as extendability during training and interpretability, as discussed in Appendix E. These benefits highlight the value of the KAN approach, which we believe justifies its inclusion in the main text.
>
> We hope this response clarifies any misunderstandings and addresses your concerns. If you have any further questions or identify any mistakes, please do not hesitate to let us know. We sincerely hope that you will reconsider and potentially increase the score for our paper.
>
> [1]Li, Z., Kovachki, N., Azizzadenesheli, K., Liu, B., Bhattacharya, K., Stuart, A., & Anandkumar, A. (2020). Fourier neural operator for parametric partial differential equations. arXiv preprint arXiv:2010.08895.
>
> [2]Li, Z., Huang, D. Z., Liu, B., & Anandkumar, A. (2023). Fourier neural operator with learned deformations for pdes on general geometries. Journal of Machine Learning Research, 24(388), 1-26.
>
> [3]Li, Z., Meidani, K., & Farimani, A. B. (2022). Transformer for partial differential equations' operator learning. arXiv preprint arXiv:2205.13671.
>
> [4]https://www.math.uni-hamburg.de/home/gunesch/calc1/chapter11.pdf (Theorem 11.13)

---

> > ### Comment · Reviewer_kDgT · 2024-08-07
> > **reply**
> >
> > Thank you for the authors' addressing to my comments and questions.
> > Other than the following two points, I've been satisfied.
> >
> > W2:  Without validation dataset, how was the test set performance measured? Early stopping? Otherwise, I slightly suspect that the result would be over-fitted to the test set, which is crucial as an ML conference paper. Please carefully validate that your results are not over-fitted to the test set, with logical explanations.
> >
> > Q4: Thank you for your explanation. Then, reading [4], I wonder what is the difference of the paper's Theorem 1 from Theorem 11.13 in [4]. Besides, is either the function space F not Hilbert space or the arbitrary function f is not at least piecewise continuous? I wonder if Theorem 1 in the paper is truly new finding or citing very classical result of Hilbert space
> > (if f is well-behaved and F is Hilbert space, any well-behaved (at least piecewise continuous) function can be expanded by an Hermitian operator's eigenfunction, which in general form a complete orthogonal basis. This is also known for more than 100 years).
> > Note that if functional space F is not Hilbert space (complete unitary space), I think the paper's theorem 1 is trying to show any orthogonal set can be complete even though in non-Hilbert space, which could be mathematically strange.
> >
> > If Theorem 1 is a really new finding, it is great. Otherwise, it just damages the reputation of the NeurIPS paper quality, in particular, from mathematics community. Please carefully specify the new point and newly reduced restriction. Although I'm not a professional researcher of Hilbert space, I can also ask for a help to my Hilbert space researcher friend.

---

> > > ### Author Response · Authors · 2024-08-08
> > > **Reply to Reviewer kDgT**
> > >
> > > Thanks for your prompt and detailed response!
> > >
> > > **W2:**
> > >
> > > Our paper follows the same evaluation methodology as the original FNO paper: using fixed training setting and training for a fixed number of epochs without early stopping. To ensure a fair comparison with other FNOs, we standardize the choice of model hyperparameters (such as the number of layers and channels) by using common settings for all FNOs including AM-FNOs. In summary, we did not use the training set to train or determine the hyperparameters. We believe that the risk of overfitting to the test set is minimal, and thus the results presented offer a fair comparison.
> > >
> > > **W3:**
> > >
> > > Sorry for the mistakes. Theorem 1 in the paper presents an alternative formulation of Theorem 11.13 in [4] from the perspective of the norm of the approximation error. The space $\mathcal{F}$ is a Hilbert space, and $f$ is any function in $\mathcal{F}$. We will revise the theorem and include the appropriate reference in the updated version. Thank you for pointing this out.
> > >
> > > If you have any further questions, we are pleased to discuss them.

---

> > > > ### Comment · Reviewer_kDgT · 2024-08-08
> > > > **reply**
> > > >
> > > > Thank you for your further explanation. However, I still need a little more information to evaluate properly.
> > > >
> > > > W2:
> > > >
> > > > Actually, I do not need an information of other papers but my question is very simple:
> > > >
> > > > "Is the reported test score evaluated at the last epoch's model weight or the best one during the training?"
> > > >
> > > > If it is the former, no problem, though I recommend to use validation set to evaluate the model's generalization ability more straight-forwardly;
> > > > If the latter, the reported result is overfitted to the test set and all the results are recommended to be re-evaluated (though it is no problem if the best performance is obtained at the last epoch for all the experiment...).
> > > >
> > > > As far as I checked the source code (naca.py / pipe.py), the test set evaluation is performed every epoch (which is not recommended on test set but only for validation set). And I cannot judge at which epoch's result is reported in the paper.
> > > >
> > > > If by accident the latter is used, then simulating early stopping from log file is necessary to be done (pick up result, e.g., at 100th epoch) where the test score is still decreasing as epochs, as a fairer comparison.
> > > >
> > > > W3:
> > > >
> > > > Thank you for your clarification. Can you please tell us how to revise it?
> > > > If the authors are still going to keep it as a theorem (with citation), the conditions should be mathematically rigorous, e.g., I'm still doubt if it is possible to prove the Theorem 1 with "arbitrary function" f in Hilbert space, which I'm not so sure if only including at least piecewise continuous function (are very mathematically exotic functions excluded?).
> > > > (I personally do not think to claim it as "theorem" because MLP function is anyway continuous function, in which case it is obvious that the function can be expanded by a set of complete orthogonal basis).

---

> ### Author Response · Authors · 2024-08-08
> **Reply to Reviewer kDgT**
>
> Thanks for your response again.
>
> **W2:**
>
> Thank you for your explanation. We reported the test score evaluated **at the last epoch** for all the models.
>
> **W3:**
>
> In our revision, we assume that $\mathcal{F}$ is a separable Hilbert space. According to Theorem 9 in [1], there exists a complete orthonormal system (orthogonal basis), which can approximate any arbitrary function $f$ within the space, as stated in Definition 11.9 and Theorem 11.13 in [2]. We will consider introducing it directly within the text in the revised version. Thank you for your suggestion.
>
> If you have any further questions, feel free to let us know.
>
> [1] https://www.math.nagoya-u.ac.jp/~richard/teaching/s2023/SML_Tue_Tai_2.pdf
>
> [2] https://www.math.uni-hamburg.de/home/gunesch/calc1/chapter11.pdf

---

> ### Comment · Reviewer_kDgT · 2024-08-08
> **reply**
>
> Thank you for your reply.
>
> W2:
>
> Please report a part of the revised Table 2 (performance at the final epoch, multi-dimensional PDEs are preferable), to assess the effectiveness of Amortized FNO in comparison to the other models again. I expect that it does not need so much effort but only checking log files...

---

> > ### Author Response · Authors · 2024-08-09
> > **Reply to Reviewer kDgT**
> >
> > Thanks for your response.
> >
> > # W2:
> >
> > Sorry for the misunderstanding. Our response above intended to clarify that we have used the final epoch performance in Table 2. Therefore, we believe that Table 2 already meets your requirement to fairly assess the effectiveness of all models.
> >
> > If you have any further questions, feel free to let us know.

---

> > > ### Comment · Reviewer_kDgT · 2024-08-09
> > > **reply**
> > >
> > > Thank you for the clarification. Now everything becomes clear for me.
> > > Although the paper should be revised with the above information before publication, I believe the author continue their hard work to polish it. So, I am willing to raise the score.
> > > Thank you for the responses.

---

> > > > ### Author Response · Authors · 2024-08-09
> > > > **Reply to Reviewer kDgT**
> > > >
> > > > Thank you very much for the detailed feedback and the improved score. We will carefully consider the points discussed and revise our paper accordingly.

---

### Official Review · Reviewer_okt7 · 2024-07-10

**Soundness:** 3
**Presentation:** 3
**Contribution:** 3
**Rating:** 7
**Confidence:** 5

**Summary:**

This paper introduces Amortized Fourier Neural Operators (AM-FNOs), a novel approach to improve Fourier Neural Operators (FNOs) for solving PDEs. The key contributions to at least me are:

1. An amortized neural parameterization of the kernel function in FNOs to accommodate arbitrarily many frequency modes using a fixed number of parameters.
2. Two implementations of AM-FNO: one based on Kolmogorov-Arnold Networks (KAN) and another using Multi-Layer Perceptrons (MLPs) with orthogonal embedding functions.
3. Theoretical analysis of the approximation capabilities of AM-FNOs.
4. Extensive empirical evaluation demonstrating significant performance improvements over existing neural operator baselines across diverse PDE benchmarks.

**Strengths:**

1. Novelty: The paper presents a novel approach to address a significant limitation of FNOs - the trade-off between model complexity and the ability to represent high-frequency details. The amortized parameterization is an innovative solution to this problem.
2. Theoretical foundation: The authors provide a solid theoretical analysis of their approach, including a theorem on the approximation properties of orthogonal basis functions. This adds depth to the empirical results and helps understand why the proposed method works.
3. Comprehensive experiments: The evaluation is thorough, covering six diverse PDE benchmarks and comparing against multiple state-of-the-art baselines. The inclusion of both in-distribution and out-of-distribution tests, as well as zero-shot super-resolution experiments, strengthens the claims of generalization ability.
4. Performance improvements: The reported improvements in accuracy (up to 35% average reduction in relative error) are substantial and consistent across different PDE types, which is impressive given the diversity of the benchmarks.
5. Ablation studies: The paper includes detailed ablation studies that provide insights into the importance of different components of the proposed method, such as the orthogonal embedding and the dimensional factorization trick.

**Weaknesses:**

1. Inadequate baseline tuning: A major weakness is that the baselines were not properly tuned. The authors use default hyperparameters or settings from previous papers for the baseline models, which may not be optimal for the specific benchmarks used in this study. This raises questions about the fairness of the comparisons and the true extent of AM-FNO's improvements.
2. Limited discussion on scalability: The paper focuses on 1D and 2D PDEs. A more in-depth discussion on the scalability of the approach to higher-dimensional problems would be valuable.
3. Computational efficiency: While the paper discusses the parameter efficiency of AM-FNOs, it doesn't provide a comprehensive analysis of the computational efficiency in terms of training time and inference speed compared to baseline methods.

**Questions:**

I just have a few, maybe it wont be answered in the rebuttal phase, but just want to know if the authors have done any studies on this:
1. How does the performance of AM-FNOs compare to multi-scale approaches like U-FNO for PDEs with significant multi-scale behavior?
2. The paper focuses on the forward problem of solving PDEs. How well might AM-FNOs perform on inverse problems or parameter estimation tasks?
3. How does the computational complexity of AM-FNOs compare to standard FNOs and other baselines, particularly for high-dimensional PDEs?
4. The paper mentions the potential of KANs for interpretability. This is evident from the original KAN paper as you can recover symbolic expressions. Could the authors elaborate on how this interpretability could be leveraged in the context of PDE solving? Would the recovered be a close enough analytic solution ( which often in PDEs we dont have any).

**Limitations:**

The authors have described their limitations!

---

> ### Author Rebuttal · Authors · 2024-08-07
>
> We sincerely thank Reviewer okt7 for recognizing the novelty and empirical contributions of our method. Below, we address the questions raised.
>
> W1: Inadequate baseline tuning.
>
> We appreciate the reviewer's concern regarding baseline tuning. To ensure a fair comparison, we have adjusted all models in Table 2 to maintain comparable parameters, with hyperparameters detailed in Appendix B. Specifically, we standardized the width and depth across all FNOs baselines (including AM-FNO) to assess the effectiveness of our amortized parameterization. Furthermore, for the critical hyperparameter of FNO, the number of retained modes, we increased it to cover all modes as demonstrated in Table 3. Even under these conditions, our methods consistently outperformed the baselines.
>
> W2: Limited discussion on scalability.
>
> Currently, our study focuses on 1D and 2D PDEs. As demonstrated in Table 3, covering all frequency modes with FNO requires nearly 9 times the number of parameters, which highlights its limitations even in 2D PDEs. We evaluate our method on a 3D benchmark, Plasticity [1], and the results, as shown in the table below, illustrate its efficiency and scalability. We will provide a more detailed discussion in the revised version.
>
> |Benchmark|AM-FNO(MLP)|AM-FNO(KAN)|Geo-FNO|
> |------|--------|---------|---------|
> |Plasticity|3.04e-3|6.11e-3|7.4e-3|
>
> W3:Computational efficiency.
>
> In the table below, we compare the training times, inference speeds, and GPU memory usage for AM-FNOs and FNO on the 2D Darcy benchmark with a resolution of $421 \times 421$. The results indicate that AM-FNO (MLP) exhibits both reduced memory usage and shorter training times compared to FNO, which is attributable to its lower complexity. Although AM-FNO (KAN) demonstrates increased training time due to its architectural design, it still benefits from lower memory consumption. We conjecture that this advantage is particularly evident when solving PDEs with high resolution and high dimensions. During inference, AM-FNOs (with kernels generated by MLP or KAN being precomputed) exhibit a similar speed to FNO, as both methods rely on similar kernel calculations. We will include a more thorough discussion in the revised version of the paper.
>
> |Model|Memory|Train Time|Inf Time|
> |-----|------|------|------|
> |AM-FNO(MLP)|**9.7G**|**43.1s**|2.4s|
> |AM-FNO(KAN)|13.5G|83.8s|2.2s|
> |FNO|14.9G|45.6s|2.2s|
>
> Q1 & Q2: Multi-scale, inverse problem and parameter estimation tasks.
>
> We have evaluated AM-FNOs on the Airfoil benchmark, which has some multi-scale features, and achieved state-of-the-art results. A more detailed assessment of AM-FNO on benchmarks with significant multi-scale behavior will be addressed in future work. Our focus is on the forward problem for PDEs. Its effectiveness on inverse problems and parameter estimation is yet to be explored and will be also considered in future research.
>
> Q3: Computational complexity.
>
> Please refer to W3.
>
> Q4: The interpretability of KAN.
>
> Thank you for your insightful question. In AM-FNO, the black-box nature of the MLP between Fourier integral operators limits interpretability. However, leveraging KANs for symbolic expression recovery is promising and could enhance interpretability in future work. This is an area we plan to explore further.
>
> We sincerely appreciate your valuable insights and corrections. We will revise our manuscript accordingly. If you have any further questions or identify any mistakes, please feel free to correct us.
>
> [1] Zongyi Li, Daniel Zhengyu Huang, Burigede Liu, and Anima Anandkumar. Fourier neural operator with learned deformations for pdes on general geometries. arXiv preprint arXiv:2207.05209,2022.

---

> > ### Comment · Reviewer_okt7 · 2024-08-09
> > **Response to authors**
> >
> > Thank you for your efforts to ensure fair comparisons by standardizing parameters across models and exploring full mode coverage are appreciated. The additional results on the 3D Plasticity benchmark demonstrate AM-FNO's scalability potential. Thanks also for including the computational efficiency comparison showing AM-FNO (MLP)'s advantages in memory usage and training time. I will not update my score however I hope that the revised version with more detailed discussions on these aspects will be included:) Thank you!

---

> > > ### Author Response · Authors · 2024-08-11
> > > **Reply to Reviewer okt7**
> > >
> > > Thank you very much for the valuable feedback. We will carefully consider the points discussed and revise our paper accordingly.

---

### Official Review · Reviewer_NuVs · 2024-07-11

**Soundness:** 2
**Presentation:** 2
**Contribution:** 2
**Rating:** 5
**Confidence:** 4

**Summary:**

Typically, FNOs require a large number of parameters when addressing high-dimensional PDEs or when a high threshold for frequency truncation is needed. To overcome this challenge, the authors introduce the Amortized Fourier Neural Operator (AM-FNO). Their method uses an amortized neural parameterization of the kernel function to handle an unlimited number of frequency modes with a fixed number of parameters. The authors provide two implementations of AM-FNO: one based on the Kolmogorov–Arnold Network (KAN) and the other using Multi-Layer Perceptrons (MLPs) with orthogonal embedding functions.

**Strengths:**

1. Improved performance: AM-FNOs, as shown by the authors, consistently achieve better performance across multiple benchmarks.

**Weaknesses:**

1. The curse of dimensionality (CoD) is not lessened by any extent. For each Fourier layer in FNO, we have to perform FFT and IFFT, which have a complexity of \( O(n \log n) \). Even if we use the full spectrum, the pointwise multiplication would only take \( O(n) \). It remains unclear to me whether it is meaningful to tackle the CoD issue in the number of parameters.
2. The proposed method seems to have some flaws. See Questions.
3. The presentation of this paper is very poor. For example, the way FNO is presented is non-standard and seems to be more from the view of the actual implementation. For example, in Section 3.2, $d_h$, correct me if I'm wrong, is the number of channels (i.e., width in FNO implementation). You should clearly state this. In line 98, $R(k): \mathcal{E} \rightarrow \mathbb{C}^{\left(d_h \times d_h\right)}$, you have $d_h$ as the number of input channels and $d_h$ as the number of output channels. That's why your codomain is $\mathbb{C}^{\left(d_h \times d_h\right)}$, and these two do not necessarily equal, although in the implementation from the FNO paper they are equal. If you choose to present FNO from an actual implementation perspective, you should clearly explain everything, especially for readers who are not familiar with the actual implementation.

**Questions:**

I'm not familiar with the Kolmogorov–Arnold Network (KAN), which is new and controversial as far as I know, so I will not comment on it.

1. It is unclear to me how this can reduce the number of parameters. Suppose for one channel in FNO, the size of the (complex) kernel is $k$. Doesn't the MLP used to generate this kernel must contain more parameters? Otherwise, let's say you want a kernel of size $k$. The output size of your MLP is $k$, then the $W$ matrix in your output layer should be of size $k' \times k$, where $k'$ is the size of the output of the previous layer.

2. Assume that you can use an MLP with fewer parameters to generate a large kernel. Doesn't this also limit the expressivity of the kernel you can learn? Essentially, what you are doing is trying to reduce the dimension of the kernel, but Fourier is already very effective in this. If your data is translationally invariant or your PDE is translationally equivariant (which is an assumption of FNO), then the optimal PCA basis consists of Fourier vectors, and you can relate this to the Eckart-Young theorem.

3. In Table 2, why is the number of parameters and running time not reported? Are the experiments repeated several times to reduce the effect of randomness? Without this information, Table 2 means nothing to me.

I apologize if the authors think my comments are too harsh. If I have indeed misunderstood certain parts of this paper, I am open-minded for a discussion and willing to adjust my views during the rebuttal.

**Limitations:**

Ｔhe authors have addressed the limitations．

---

> ### Author Rebuttal · Authors · 2024-08-07
>
> We thank Reviewer NuVs for the valuable feedback. Below, we respond to the questions.
>
> W1: The reason for tackling the CoD issue in the number of parameters.
>
> In FNO, the kernel is parameterized independently for each frequency mode, resulting in complexity for the Fourier integral operator of $O((d_h^2 k)^{D}))$, where $d_h$ represents both the input and output channel dimensions, $k$ denotes the number of retained modes, and D indicates the spatial dimensionality. For high-resolution and high-dimensional data, this leads to significant memory consumption, as a large number of modes need to be retained. For example, one 64-width FNO layer for 2D PDEs can exceed 130 million parameters with $k=256$. This severely limits practical applications, such as large-scale pre-trained models.
>
> We provide a comparison of the training time and memory consumption for AM-FNOs and FNO ( with all modes retained) in the table below, tested on the 2D Darcy benchmark with a resolution of $421 \times 421$. The results indicate that AM-FNO (MLP) exhibits both reduced memory usage and shorter training times compared to FNO, which is attributable to its lower complexity. Although AM-FNO (KAN) demonstrates increased training time due to its architectural design, it still benefits from lower memory consumption. We conjecture that this advantage is particularly evident when solving PDEs with high resolution and high dimensions, attributed to AM-FNO's reduced complexity. A comprehensive analysis will be included in the updated version of the paper.
>
> |Model|Memory|Train Time|
> |-----|------|------|
> |AM-FNO(MLP)|**9.7G**|**43.1s**|
> |AM-FNO(KAN)|13.5G|83.8s|
> |FNO|14.9G|45.6s|
>
> W2: Paper presentation.
>
> Thank you for your feedback on the presentation. We apologize for any confusion caused by the brief introduction of FNO due to page limitations. We will provide a clearer explanation in the updated version to ensure understanding, especially for readers not familiar with the implementation details.
>
> Q1: Parameter reduction
>
> We would like to clarify that in AM-FNO, the MLP is used to map each frequency mode (after embedding) to its corresponding kernel value, meaning the number of parameters depends on the number of channels rather than the number of retained modes. For instance, if an MLP with $m$ basis functions processes input and outputs a kernel of size $k$, the parameter count is approximately $2m^2 + 2m$ (assuming one hidden layer and one channel). In contrast, FNO requires $k$ parameters for the kernel, which can be expensive for high-resolution and high-dimensional data (more modes should be retained).
>
> Q2: MLP limits the expressivity.
>
> While Fourier methods are theoretically efficient, they require a large number of parameters when handling high-dimensional and high-resolution data, which can be computationally prohibitive. Using an MLP to generate the kernel might indeed limit expressivity compared to parameterizing every frequency mode as in FNO theoretically. We regard this as a tradeoff. However, our empirical results demonstrate that AM-FNO outperforms FNO covering all frequency modes (see Table 3). We believe this is because the smoother transformations facilitated by the MLP improve optimization efficiency, leading to better performance.
>
> Q3: Main experiments.
>
> Many results in Table 2 are sourced from the original papers, making a direct comparison of running times potentially unfair due to hardware differences. All the baselines in Table 2 have comparable numbers of parameters, with their hyperparameters detailed in Appendix B. Below, we provide a parameter comparison on Darcy benchmark to validate this. The results of neural operators are generally consistent despite randomness, which is why many popular baselines do not report repeated results [1,2]. We repeated our experiments three times on CFD-2D and NS-2D benchmarks, as shown in the table below, demonstrating the stability of our method.
>
> | |AM-FNO(MLP)|AM-FNO(KAN)|FNO|U-FNO|OFormer|LSM|
> |------|--------|---------|------|-------|---------|--------|
> |Params.(M)|1.1|1.5|2.4|1.3|2.7|4.8|
>
> |Benchmark|AM-FNO(MLP)|AM-FNO(KAN)|
> |------|--------|---------|
> |NS-2D|8.53e-2 ± 7.48e-4|1.04e-1 ± 3.27e-3|
> |CFD-2D|2.21e-3 ± 4.13e-5|2.75e-3 ± 8.96e-5|
>
> Thank you for your openness and willingness to discuss further. We hope this discussion clarifies any misunderstandings and addresses your concerns. If you have any further questions or identify any mistakes, please do not hesitate to let us know.
>
> [1] Li, Z., Kovachki, N., Azizzadenesheli, K., Liu, B., Bhattacharya, K., Stuart, A., & Anandkumar, A. (2020). Fourier neural operator for parametric partial differential equations. arXiv preprint arXiv:2010.08895.
>
> [2] Hao, Z., Wang, Z., Su, H., Ying, C., Dong, Y., Liu, S., ... & Zhu, J. (2023, July). Gnot: A general neural operator transformer for operator learning. In International Conference on Machine Learning (pp. 12556-12569). PMLR.

---

> > ### Comment · Reviewer_NuVs · 2024-08-07
> > **Reply to Author's Rebuttal**
> >
> > ### W1
> > > This severely limits practical applications, such as large-scale pre-trained models.
> >
> > First of all, how do you get $\left.O\left(\left(d_h^2 k\right)^D\right)\right)$? Isn't it just $\left.O\left(d_h^2 k^D\right)\right.$
> >
> > Can you provide a case in which they use large pre-trained FNOs?
> >
> > Moreover, in your case, FFT would take $O(d_h n \log n)$, where $n \geq k^D$. FFT itself is already very expensive and suffers from the CoD issue.
> >
> > ### Q1
> > Can you be more specific, such as providing some mathematical expressions? I'm still confused.
> >
> > ### Q3
> > I do understand that the rebuttal period is only one week, and you might not have time to do this.  However, I believe that running times are important. A simple example is that a U-Net or DeepONet with a similar number of parameters might be much faster than FNO. If you run their models and record the time on the same machine and with the same settings, I believe you can ensure the comparison is consistent and fair.

---

> ### Author Response · Authors · 2024-08-08
> **Reply to Reviewer NuVs**
>
> Thanks for your prompt and detailed response!
>
> **W1:**
>
> We appreciate your feedback and would like to address your concerns subsequently.
>
> Firstly, we apologize for the error in the complexity statement; it should be $O(d_h^2k^{D})$.
>
> Secondly, there have been attempts to utilize large pre-trained FNO models. For instance, DPOT [1] explores a model that employs a shared MLP to transform each frequency mode of the input (akin to a convnet with a $1 \times 1$ kernel) to mitigate the memory overhead of kernels. This architecture is similar to AFNO [2], which we have compared in Table B of the global rebuttal. Our results indicate that our method outperforms AFNO on both benchmarks, which we attribute to AFNO’s limited expressiveness due to its uniform transformation across different frequency modes, while our approach treats each mode differently.
>
> Regarding your concerns about FFT, we acknowledge that FFT suffers from the Curse of Dimensionality (CoD). However, FNOs still demonstrate superior training speed (as shown in the table below in Q3) and prediction accuracy compared to other neural operators. Our method primarily addresses memory issues, particularly in high-resolution or high-dimensional contexts, as demonstrated in the table in the rebuttal before. Our approach shows a reduction in memory usage (and shorter training time for AM-FNO (MLP)) even in 2D benchmarks, suggesting potential benefits for large-scale models.
>
> **Q1:**
>
> Continuing with the assumption of one channel and a one-dimensional kernel size of $k$, let’s further simplify the MLP to a linear layer for clarity. To map frequency modes (with shape $[k,1]$ ) to the corresponding kernel values (also with shape $[k,1]$ ), our method first applies basis functions to obtain an embedding with shape $[k,m]$ , where m is the number of basis functions. We then compute the kernel values (real or imaginary part) using a linear layer with $m \times 1$ parameters. Consequently, the complexity of our method depends only on the number of channels and the number of basis functions (specifically $O(D m d_h^2)$ in the linear case compared to $O(d_h^2k^{D})$ in FNO for multi-dimensional PDEs), and avoid becoming excessively large with increasing dimensions and resolution.
>
> **Q3:**
>
> To address your concern, we provide a comparison of training times under our experimental settings for the same benchmark, as shown in the table below. The results indicate that our method has a slower training speed compared to other FNOs, which can be attributed to the additional frequency modes we adopted. However, our method still outperforms non-FNO neural operators. Furthermore, as shown in the previous rebuttal, when all modes are retained, our method can surpass FNO in training time.
>
> | |AM-FNO(MLP)|AM-FNO(KAN)|FNO|U-FNO|Oformer|LSM|
> |-----|-----|-----|-----|-----|-----|-----|
> |Train Time (s/epoch)| 1.2 | 2.1 | 0.9 | 1.6 | 16.5 | 2.1 |
>
> If you have any further questions, we are pleased to discuss them.
>
> [1] Hao, Z., Su, C., Liu, S., Berner, J., Ying, C., Su, H., ... & Zhu, J. (2024). Dpot: Auto-regressive denoising operator transformer for large-scale pde pre-training. arXiv preprint arXiv:2403.03542.
>
> [2] Guibas, J., Mardani, M., Li, Z., Tao, A., Anandkumar, A., & Catanzaro, B. (2021). Adaptive fourier neural operators: Efficient token mixers for transformers. arXiv preprint arXiv:2111.13587.

---

> > ### Comment · Reviewer_NuVs · 2024-08-09
> >
> > Overall, I appreaciate your efforts to address my concerns; I will be satisfied if the following points can be addressed/clarified:
> >
> > >　Firstly, we apologize for the error in the complexity statement; it should be $O\left(d_h^2 k^D\right)$.
> >
> > My main concern is that while your work aims to reduce the curse of dimensionality in FNO, but the complexity introduced by FFT might undermine these benefits from your method.
> >
> > > Our method primarily addresses memory issues, particularly in high-resolution or high-dimensional contexts.
> >
> > Does your method address the time complexity issue in these scenarios? From what I understand, your approach might actually increase the running or inference time. The primary reason for using neural operators is to achieve fast inference; otherwise, numerical schemes with coarse discretization can offer even better performance with theoretical guarantees.
> >
> > > Q1
> >
> > So, are the MLPs the same for every dimension? What happens if the axes aren't on the same scale? For instance, in a 2D domain of $[0, 1000] \times [0, 1]$, where you want to keep 100 modes for the first dimension and only 10 for the second, can your method still work given that MLPs are fixed in size? If not, it seems your method can only be applied when you sample the same number of frequency modes (i.e. $k$) in every dimension, which doesn’t seem practical to me.
> >
> > For some test data, e.g., from the FNO paper, you can effectively do this. But not pratical in general.
> >
> > > However, our method still outperforms non-FNO neural operators.
> >
> > There are many non-FNO neural operators, e.g., DeepONet, U-Net, SNO [1], the selected baselines such as Oformer and LSM is not what I would have expected to see as a baseline in this venue.
> >
> > [1] Spectral Neural Operators, V. Fanaskov, I. Oseledets

---

> > > ### Author Response · Authors · 2024-08-11
> > > **Reply to reviewer Reviewer NuVs**
> > >
> > > Thanks for your feedback.
> > >
> > > 1) We understand your reasonable concerns about the complexity of FFT; however, FNOs still demonstrate superior speed compared to other neural operators. We will include a discussion on the limitations of FFT in the revised version.
> > >
> > > 2) While our models require more training time than the standard FNO due to the use of MLPs or KANs, the difference is relatively minor, as shown in the table presented before(1.2s for AM-FNO (MLP) vs. 0.9s for FNO). In terms of inference time, we can precompute the kernel using the trained MLPs or KANs before inference, resulting in the same computational complexity as the FNO.
> > >
> > > 3) The MLP is different for every dimension. We present the result on 3D Plasticity benchmark, where we kept 101, 31, 20 modes for every dimension. As shown, our method outperform Geo-FNO.
> > >
> > > | Benchmark  | AM-FNO(MLP)  | Geo-FNO |
> > > | ---------- | -----------  | ------- |
> > > | Plasticity | 3.04e-3      | 7.4e-3  |
> > >
> > > 4) Thanks for your suggestion. We will add the mentioned baselines in the revised version.
> > >
> > > If you have any further questions, we are pleased to discuss them.

---

> > > > ### Comment · Reviewer_NuVs · 2024-08-12
> > > >
> > > > > We understand your reasonable concerns about the complexity of FFT; however, FNOs still demonstrate superior speed compared to other neural operators. We will include a discussion on the limitations of FFT in the revised version.
> > > >
> > > > FNO does not always demonstrate superior speed. How can you fairly compare the speed of FNO with that of DeepONet or UNets?
> > > >
> > > > Moreover, since it performs convolution, i.e., pointwise multiplication in the spectral domain? The complexity of the number of parameters or FLOPs is just $O(n)$, while FFT takes $O(n\log n)$, where $n$ is the number of sensor points. I feel this is an important aspect to think about when you think of the complexity of the number of parameters.
> > > >
> > > > > While our models require more training time than the standard FNO due to the use of MLPs or KANs, the difference is relatively minor, as shown in the table presented before(1.2s for AM-FNO (MLP) vs. 0.9s for FNO). In terms of inference time, we can precompute the kernel using the trained MLPs or KANs before inference, resulting in the same computational complexity as the FNO.
> > > >
> > > > Ok, this makes sense to me. You should include this in the paper as part of the discussion.
> > > >
> > > > > The MLP is different for every dimension. We present the result on 3D Plasticity benchmark, where we kept 101, 31, 20 modes for every dimension. As shown, our method outperform Geo-FNO.
> > > >
> > > > I see. However, can you clarify why you chose Geo-FNO as a baseline? Geo-FNO tackles non-rectangular geometries. If your test data is rectangular, then why use Geo-FNO? If it's not, then how do FNO, let alone AM-FNO, even work?
> > > >
> > > >
> > > > Since some concerns and questions have been addresed, I have changed my rating.

---

> ### Author Response · Authors · 2024-08-13
> **Reply to Reviewer NuVs**
>
> Thanks for your feedback and improved score. Below, we further respond to your questions.
>
> 1） Due to the time constraints of Rebuttal, we directly used the official Geo-FNO code to compare the speed of Geo-FNO and DeepONet [1]. Note that DeepONet requires function values at all input points and a query coordinate but outputs only the value at the coordinate, while FNO outputs values for all points. To align batch sizes, we used a batch size of $16$ for Geo-FNO and $16 \times 972$ (972 is the number of sensor points) for DeepONet. As shown in the table below, huge batch sizes for DeepONet can pose significant challenges for data transfer and parallel processing, resulting in reduced efficiency. While FFT may become a limitation in very high-dimensional settings, FNO still outperforms in this context. We also compare FLOPs for computing an entire function (972 points) to better illustrate the efficiency of the two models.
>
> | Model    |   Params.(M) | Train Time (s/epoch) |   Inf Time (s/epoch)   | FLOPs（Calculation for 972 Points） |
> |----------| ---------|---------| ------- |------- |
> | DeepONet | 1.0 | 13.9 | 2.5 | 0.99B |
> | Geo-FNO  | 1.5 | 2.1 | 0.2 | 0.11B |
>
> Regarding U-Net, while it may be more efficient in many scenarios, its local convolution in the spatial domain limits performance when testing across different discretizations, which is significant for neural operators [2].
>
> Thank you for your questions on efficiency. We recognize that the complexity of FFT may pose a limitation for FNOs and will include a detailed discussion on COD in our revised version.
>
> 2）The Plasticity benchmark is presented on a structured mesh, where the indexing induces a canonical coordinate map and enables the direct application of FFT  without the learnable mapping [3]. Consequently, Geo-FNO is equivalent to the standard FNO in this context. We refer to it as Geo-FNO to maintain consistency with the original Geo-FNO paper. Thank you for pointing this out.
>
> We hope the above response can further address your concern. If you have any further questions, we are pleased to discuss them.
>
> [1] https://github.com/neuraloperator/Geo-FNO
>
> [2] Wen, G., Li, Z., Azizzadenesheli, K., Anandkumar, A., & Benson, S. M. (2022). U-FNO—An enhanced Fourier neural operator-based deep-learning model for multiphase flow. Advances in Water Resources, 163, 104180.
>
> [3] Li, Z., Huang, D. Z., Liu, B., & Anandkumar, A. (2023). Fourier neural operator with learned deformations for pdes on general geometries. Journal of Machine Learning Research, 24(388), 1-26.

---

> > ### Comment · Reviewer_NuVs · 2024-08-13
> >
> > > Geo-FNO is equivalent to the standard FNO in this context.
> >
> > Your aim is to resolve or lessen the CoD issues in FNO. It would be more interesting to see a comparison between a large FNO and a large AM-FNO regarding the improvement in 1) memory usage, 2) training and inference time, and 3) FLOPs. However, I understand that due to the limited time available during the rebuttal, it may be almost impossible to obtain such results.
> >
> > >  While FFT may become a limitation in very high-dimensional settings, FNO still outperforms in this context.
> >
> > In Section 5.1, you mentioned DeepONet as a baseline, but I do not see it in the results (Table 2 or any other tables). I'm trying to find the L2 error information on this. Moreover, in my experience, DeepONet is usually faster than an FNO model with a similar number of parameters on normal uniform rectangular domain data (implementation follows directly from [1], a torch adaptation can also be found from the implemetation of [2]). However, I do understand that implementation details and hardware details may differ.
> >
> >
> > Given that most of my concerns and questions have been addressed, and recognizing the authors' efforts to include additional results within the short rebuttal period, I am willing to increase my score. However, since my major concern remains unresolved and the authors agree it should be acknowledged as a limitation, I will not be making any further adjustments to the score.
> >
> > [1] A comprehensive and fair comparison of two neural operators (with practical extensions) based on FAIR data
> >
> > [2] Physics-Informed Neural Operator for Learning Partial Differential Equations

---

> ### Author Response · Authors · 2024-08-13
> **Reply to Reviewer NuVs**
>
> Thank you for making such a constructive discussion with us!
>
> We totally understand your concerns about method efficiency and have tried our best to demonstrate that our approach may not suffer from issues from that aspect during the rebuttal. Of course, we will add the mentioned comparison between large FNO and large AM-FNO in the revision. Additionally, we will provide a comprehensive comparison with DeepONet, particularly focusing on efficiency.
>
> At last, we also clarify that alleviating the parameter count issue is a side product of this paper. A more evident effect of the proposed amortized parameterization is to foster the frequency modes to communicate with each other, leading to enhanced predictive performance. We will carefully revise the paper to weaken the argument on addressing COD in our revised version.
>
> Thanks again!

---

### Official Review · Reviewer_Z566 · 2024-07-11

**Soundness:** 2
**Presentation:** 2
**Contribution:** 2
**Rating:** 5
**Confidence:** 4

**Summary:**

This paper presents the AMortized Fourier Neural Operator (AM-FNO), which utilizes an amortized neural representation of the kernel function. It allows accommodating a variable number of frequency modes while using a fixed number of parameters compared to the Vanilla Fourier Neural Network.

**Strengths:**

S1) Amortized neural parameterization of the kernel function using MLP and KAN amortized neural parameterization of the kernel function.

S2) The approach explores high-frequency components without increasing the parameters of the Fourier Neural Operator.

**Weaknesses:**

W1) Baselines seem limited, and GNOT, Transsolver, ONO, UNET, etc., need to be included.

W2) The benchmarks have been tested consistently on a structured grid but have yet to be tested on an unstructured mesh.

W3) The proposed approach resembles the spectral neural operator which uses the Fourier and Chebyshev series,

W4) Using low-rank approximation to approximate the Fourier kernel or incorporating depthwise convolution with non-linearity in baselines will enhance comprehension of the proposed method.

**Questions:**

Q1) What is your rationale for employing an orthogonal basis? Have you experimented with applying depthwise convolution in FFT space, followed by non-linear transformations and another round of depthwise convolution?

Q2) Could you please clarify why KAN consistently performs worse than MLP? Given that the dataset is noise-free, shouldn't KAN be expected to outperform MLP?

Q3) The FFNO number reported does not correspond to the number in the original paper.

Q4) Have you attempted using a low-rank approximation of the kernel weight in Fourier space within FNO?

Q5) Difference between the proposed method and spectral neural operator.

Q6) Why does having orthogonal embedding assist in AM-FNO? Have you experimented with employing the standard FFN directly? What type of orthogonal embeddings were used for the experiment?

**Limitations:**

Yes, the authors adequately addressed the limitations.

---

> ### Author Rebuttal · Authors · 2024-08-07
>
> We thank Reviewer Z566 for the valuable feedback. Below, we respond to the questions. **Please note that the additional tables and references are included in the Global Rebuttal due to character limits.**
>
> W1: More baselines.
>
> Our method aims to enhance Fourier neural operators (FNOs), a significant subclass of neural operators. Therefore, we primarily use FNOs as our baselines, and our method demonstrates superior performance. Additionally, we have included widely used baselines (OFormer) and a competitive baseline (LSM) to validate our method's effectiveness. In response to your suggestion, we include some mentioned baselines, as shown in Table B. It indicates that our method outperforms the additional baselines as well. We will include these results in the revised version.
>
> W2: Benchmarks on unstructured mesh.
>
> Thanks for your suggestion. We evaluate our method on the Elasticity benchmark presented on point clouds [1] in Table C. To handle irregular geometry, we implemented the widely used Geo-FNO method to map the irregular meshes to and from uniform meshes [1]. Our results show that our method outperforms other neural operators that utilize the same learnable mapping. However, it performs worse than transformer-based neural operators, which naturally process input functions on irregular geometry as a sequence. We hypothesize that the learnable mapping introduces errors. We will include these results in the revised version.
>
> W3: Spectral Neural Operator Resemblance.
>
> We would like to clarify that our method is **fundamentally different from SNO**. SNO utilizes Chebyshev polynomials to represent functions with finite sets of coefficients and learns the mapping between these coefficients. This results in band-limited operators that are restricted to generating frequencies within a fixed representation space.
> In contrast, FNOs learn the direct mapping between functions and compute the integral operator in Fourier space for high inference speed. Empirically, FNOs demonstrate the ability to extrapolate to higher frequencies not seen during training, which SNOs lack due to their fixed representation limits [2]. As a variant of FNO, while AM-FNO employs orthogonal basis functions to approximate the kernel function—a common practice in function approximation—the neural operator itself is fundamentally distinct from SNO.
>
> W4: Low-rank approximation or depthwise convolution with non-linearity.
>
> FNO with low-rank approximation is similar to one of our baselines, F-FNO, which factorizes the kernel across different dimensions. The results in Table 2 (**in the paper**) demonstrate that AM-FNO outperforms F-FNO, highlighting the superiority of our amortized parameterization over the point-by-point parameterization in F-FNO.
>
> Regarding the non-linear convolution, AFNO [3] employs a non-linear convnet to evolve features at every frequency mode. We evaluate AFNO on two benchmarks, and the results, as shown in Table B, indicate that AFNO has lower accuracy. We hypothesize that this is because AFNO uses the same MLP to evolve all frequency modes of the input, potentially limiting expressiveness. In contrast, AM-FNO transforms different frequency modes separately. We will include further discussion on AFNO in the revised version.
>
> Q1: Our rationale for employing orthogonal basis functions stems from the observed performance degradation when using an MLP to directly approximate the mapping between frequencies and kernel values, as shown in Table 4 (**in the paper**). The MLP struggles to capture the complex, non-linear nature of the kernel function. Embedding scalar inputs with predefined functions is a well-established technique in machine learning, as exemplified by time embeddings in diffusion models. This approach helps MLPs convert linear inputs into expressive, non-linear forms effectively. Inspired by it, we employ orthogonal functions to embed the frequencies, mapping them into a high-dimensional and non-linear feature space. This introduces an inductive bias that enhances the efficiency of approximating kernel functions. Regarding the depthwise convolution in FFT space, please refer to our response in W4.
>
> Q2: As reported in [4], KAN does not consistently outperform MLP. As shown in Table 4 (**in the paper**), AM-FNO with KAN outperforms AM-FNO using MLP directly and performs worse than AM-FNO using orthogonal basis functions to embed the input. This indicates the effectiveness of our embedding technique. Meanwhile, KAN has the advantage of being extendable during training by increasing the number of local basis functions. For a fair comparison, we kept the parameter count constant and did not extend KAN during training. Given the results in Figure 4 (**in the paper**), it is plausible that AM-FNO (KAN) can outperform if we increase the number of basis functions during training.
>
> Q3: The F-FNO results were reproduced using the training settings and similar hyperparameters as other FNOs, as described in Section 5.1 and Appendix B, to ensure a fair comparison. The discrepancy with the original paper's results may stem from differences in training techniques. For instance, the original F-FNO paper employs methods like enforcing the first-order Markov property and adding Gaussian noise. Our training settings align with standard practices in this domain to maintain consistency across evaluations.
>
> Q4: Please refer to W4.
>
> Q5: Please refer to W3.
>
> Q6: Please refer to Q1 and Q2. We primarily used Chebyshev basis functions for the orthogonal embeddings in our experiments. In our ablation study, we also replaced them with triangular basis functions and non-orthogonal basis functions to evaluate their impact.
>
> We hope this response clarifies any misunderstandings and addresses your concerns. If you have any further questions or identify any mistakes, please do not hesitate to let us know. We sincerely hope that you will reconsider and potentially increase the score for our paper.

---

> ### Comment · Reviewer_Z566 · 2024-08-10
>
> Thank you for the response.
> 1) The paper references factorization techniques, but the implementation uses a Chebyshev basis to parametrize the entire Fourier kernel, along with an MLP. As reviewer DLAH noted, this approach is not clearly explained in the current version of the paper, where only smooth function parametrization is mentioned. It is essential to explicitly include this discussion in the paper for clarity and the discussion about the spectral bias in MLP.
>
> 2) I couldn't find hyperparameter details for the benchmark datasets in the paper, which is needed to ensure reproducibility.
>
> 3) If you are using all Fourier frequency modes in the proposed method, then it's not a fair comparison. I would like to see the performance compared with FNO, where we have used all the frequency modes. Also, it would be great if the author could provide the performance of the proposed method using only the same number of modes as used for FNO.
>
> 4) I could find two Tx and Ty in the code. Could you clarify:
>  >     self.Tx = torch.zeros(self.n1, H+padding)
>         self.Ty = torch.zeros(self.n2, (W+padding)//2+1)
>         self.Tx = (torch.cos(self.grade1@torch.acos(self.gridx))).reshape(1, self.n1, H+padding, 1).cuda()
>         self.Ty = (torch.cos(self.grade2@torch.acos(self.gridy))).reshape(1, self.n2, 1, (W+padding)//2+1).cuda()
>
> 5) Could you compare the proposed method with baselines regarding training time, GPU consumption, inference time, and number of parameters?
>
> PS: I am open to discussing the paper and would like to consider increasing the score if my questions are addressed.

---

> ### Author Response · Authors · 2024-08-10
> **Reply to Reviewer Z566**
>
> Thanks for your feedback.
>
> 1) Thanks for your suggestion. We agree that the concept of spectral bias provides valuable insight into our approach, and we will include additional discussion to elaborate on this in the revised version.
>
> 2) Sorry for not including this part. We will include the relevant descriptions of the benchmarks in the revised version.
>
> 3) In our main experiment (Table 2), we aimed to ensure that all models had a comparable number of parameters. However, the parameter count for FNO with all modes becomes excessively large, making it unfair to compare with other models. Meanwhile, one advantage of our method is that it can capture all modes with a limited number of parameters. We have presented the results of FNO with full modes on CFD-2D (denoted as "FNO+" in Table 3), where AM-FNOs outperform FNO+. We also provide comparisons with FNO+ on the Darcy and Airfoil benchmarks below. We will include additional empirical results on FNO+ and AM-FNOs with the same number of modes in the revised version.
>
> | Model | AM-FNO (MLP) | AM-FNO (KAN) | FNO+ |
> |------|------|------|------|
> |Darcy| 4.21e-3 | 4.28e-3 | 1.33e-3 |
> | Airfoil | 5.64e-3 | 6.06e-3 | 1.32e-2 |
>
> 4) The first two lines of code are for initialization (though they aren’t necessary), while the next two lines compute the basis functions. We want to clarify that our code is consistent with our method.
>
> 5) We provide a comparison of all models, using the hyperparameters from the main experiment, tested on the same benchmark in the table below. Our methods require more memory and training/inference time compared to other FNOs, due to the additional modes we maintain. However, AM-FNOs still outperform non-FNO neural operators.
>
>
> | Model| AM-FNO (MLP)| AM-FNO (KAN) | FNO | U-FNO | OFormer | LSM | F-FNO |
> |------|------|------|------|------|------|------|------|
> |Train Time (s) |  1.2  | 2.1  | 0.9   |  1.6  | 16.5 |   2.1 |  1.1  |
> |Inf Time (s) |  0.091  |  0.17  |  0.0068  |  0.087  |  1.5 |  0.15  |  0.075  |
> |Memory (M) | 1850 |  2066  |  1212  | 1444 | 16090   |  1894  | 1126  |
> |Params. (M) |  1.1  |  1.5  |  2.4  |  2.6  |  1.3  |  4.8  |  0.2 |
>
> If you have any further questions, we are pleased to discuss them.

---

> ### Comment · Reviewer_Z566 · 2024-08-11
>
> Thank you for the response.
>
> Q2) It would be better if you could also report hyperparameters for the proposed method.
>
> Q3) With increased modes, FNO is computationally heavy and might sometimes underperform. However, can you report the numbers of AM-FNO compared with FNO having the same modes? It will help to understand the proposed method better, whether increasing the modes is effective in AM-FNO or the way in which the kernel is parametrized (Kernel Bias)
>
> Q) Can you report the numbers using just one layer of AM-FNO and FNO on specific benchmarks?
>
> I am trying to understand the proposed method better from the point where kernel bias is more critical or increasing the modes. Also, it's fine if the proposed method is not competitive with FNO in terms of computational and time complexity, as the proposed method is trying to address different problems altogether.

---

> ### Author Response · Authors · 2024-08-11
> **Reply to Reviewer Z566**
>
> Thanks for your feedback.
>
> Q2) We have reported the essential hyperparameters of our models in Section 5.1. We will provide a more detailed description in our revised version.
>
> Q3) We present the performance of our models with 12 modes compared to FNO with the same number of modes on Darcy benchmark, as shown in the table below. The results indicate that our models continue to significantly outperform the FNO, demonstrating the effectiveness of our parameterization.
>
> | Benchmark  | AM-FNO(MLP)  | AM-FNO(KAN) |   FNO   |
> |----------| ---------|---------| ------- |
> | Darcy      | 4.72e-3 | 4.78e-3 | 1.08e-2 |
>
> Q) The results of AM-FNOs and FNO with one layer on Darcy are shown below.
>
> | Benchmark  | AM-FNO(MLP)  | AM-FNO(KAN) |   FNO   |
> |----------| ---------|---------| ------- |
> | Darcy      | 1.85e-2 | 1.98e-2 | 2.17e-2 |
>
> If you have any further questions, we are pleased to discuss them.

---

> > ### Comment · Reviewer_Z566 · 2024-08-11
> >
> > Thanks for addressing my concerns. It seems both MLP inductive bias and adding modes are boosting the performance of AM-FNO. After reviewing the global and each reviewer's question responses, I have decided to raise my score based on new experimental results and discussion. I have changed my score from 4 to 5 and hope to see the discussion and all the new results in a revised version of the paper.

---

> > > ### Author Response · Authors · 2024-08-11
> > > **Reply to Reviewer Z566**
> > >
> > > Thank you very much for the valuable feedback and improved score. We will carefully consider your suggestions and revise our paper accordingly.

---

### Official Review · Reviewer_DLAH · 2024-07-12

**Soundness:** 2
**Presentation:** 3
**Contribution:** 2
**Rating:** 6
**Confidence:** 4

**Summary:**

The paper introduces the AM-FNO to address high-frequency truncation in the original FNO, which can damage the performance for PDE data with substantial high-frequency information. AM-FNO utilizes MLP or KAN to approximate the kernel function value in Fourier space for all frequency modes. For MLP based AM-FNO, orthogonal embedding functions are applied to enhance MLP's performance. Factorization are applied to reduce the total number of basis functions. Experiments on various PDE datasets show that AM-FNO consistently outperforms baseline models. The efficacy of different components in AM-FNO is valedated through ablation experiments.

**Strengths:**

1. The paper presents an original and intuitive method using MLP or KAN to approximate the kernel function values in Fourier space, contrasting with the original FNO's approach that requires separate linear functions for each frequency mode, which often leads to a high parameter count in high-dimensional PDEs.

2. The manuscript is overall well-written, with clear descriptions of the methods used.

3. AM-FNO has been tested across multiple datasets and consistently outperforms various baselines. Detailed abltation experiments are provided to validate each component's contribution.

**Weaknesses:**

1. It seems that orthogonal basis functions play a crucial role in boosting AM-FNO's performance. According to Table 4, MLP-based AM-FNO without these functions might even perform worse than the standard FNO. However, the use of orthogonal basis functions in the design is not convincingly motivated. The justification provided—that 'vanilla MLPs lack effective inductive bias for function approximation' (line 135)—is ambiguous. This explanation leaves the impression that orthogonal basis functions are used as an arbitrarily trick to enhance MLP's performance.

2. Using Fourier series as latent features is a widely used method to improve MLP's convergence [1, 2]. The use of orthogonal embedding functions in AM-FNO is very close to how Fourier features work. It's better to discuss this common Fourier features method in the paper and test it performance following the setting of Table 4.

3. The discussion and evaluation lack consideration of AFNO [3], a closely related baseline. AFNO features a key design of weight sharing, where a single MLP approximates the kernel function for all frequency modes in Fourier space. While AFNO's MLP does not take frequency mode as input like AM-FNO does, its motivation and implementation are quite similar to those of AM-FNO.


[1] Tancik, M., Srinivasan, P., Mildenhall, B., Fridovich-Keil, S., Raghavan, N., Singhal, U., ... & Ng, R. (2020). Fourier features let networks learn high frequency functions in low dimensional domains. Advances in neural information processing systems, 33, 7537-7547.

[2] Mildenhall, B., Srinivasan, P. P., Tancik, M., Barron, J. T., Ramamoorthi, R., & Ng, R. (2021). Nerf: Representing scenes as neural radiance fields for view synthesis. Communications of the ACM, 65(1), 99-106.

[3] Guibas, J., Mardani, M., Li, Z., Tao, A., Anandkumar, A., & Catanzaro, B. (2021). Adaptive fourier neural operators: Efficient token mixers for transformers. arXiv preprint arXiv:2111.13587.

**Questions:**

1. The paper states in line 110 that 'using an NN guarantees the kernel function to evolve more smoothly as the frequency mode changes, due to the smoothness of NN'. However, given the well-known low-frequency bias of MLPs [4], this smoothness might actually be the reason for MLP's suboptimal performance without orthogonal basis functions. A possible explanation for the success of orthogonal basis functions is that they introduce higher frequency components into the MLP’s input, similar to Fourier features, thereby reducing the MLP's smoothness. Could the authors clarify whether a smoother or less smooth MLP is preferable in this context?

2. Taking a 2D PDE as an example, a real-valued function transforms into a centrally symmetric complex function in Fourier space. Does the NN in AM-FNO ensure this central symmetry, such that inputs k_x,k_y produce the same output as -k_x, -k_y?

3. Does AM-FNO require more GPU memory or more training time compared to the original FNO? (It appears that the Vanilla in Table 4 is not the same as the original FNO in Table 2, given the differences in their reported errors.)

[4] Rahaman, N., Baratin, A., Arpit, D., Draxler, F., Lin, M., Hamprecht, F., ... & Courville, A. (2019, May). On the spectral bias of neural networks. In International conference on machine learning (pp. 5301-5310). PMLR.

**Limitations:**

The authors addressed the limitations of the AM-FNO method.

No potential negative societal impact is noted.

To enhance the paper, please consider address the outlined weaknesses and questions, particularly the role of orthogonal basis functions. Given their significant benefits, a more detailed analysis explaining why these functions are effective would strengthen the paper.

---

> ### Author Rebuttal · Authors · 2024-08-07
>
> We thank Reviewer DLAH for the acknowledgment of our method and empirical contributions. Below, we respond to the questions.
>
> W1: The motivation for orthogonal basis functions.
>
> Sorry for the lack of clarity. We make the following clarifications. As shown in Table 4, the performance degrades compared to the version with KAN when using an MLP to directly approximate the mapping between frequencies and kernel values. We attribute this degradation to the MLP's difficulty in capturing the complex, non-linear nature of the kernel function. Embedding scalar inputs with predefined functions is a well-established technique in machine learning, as exemplified by time embeddings in diffusion models. This approach allows MLPs to more effectively transform linear inputs into expressive, non-linear representations. Inspired by this, we employ orthogonal functions to embed the frequencies, mapping them into a high-dimensional and non-linear feature space. This method introduces an inductive bias that enhances the efficiency of approximating kernel functions. This method is supported by the theory of orthogonal functions and empirical results and has been studied in function approximation [1]. Further discussion will be provided in the revised version. Thanks for your question.
>
> W2: Discussion about the Fourier series.
>
> We experimented with triangular basis functions (TBF), which are used as basis functions in the Fourier series, as shown in Table 4. While TBF yielded better results for the Darcy benchmark, they performed worse compared to Chebyshev basis functions in other cases. We hypothesize that TBFs better capture the periodic structure of the Darcy benchmark and will discuss this further in the revised version.
>
> W3: Discussion about AFNO.
>
> Thanks for your suggestion. We evaluated AFNO on two benchmarks, as shown in the table below. The results indicate that AFNO has lower prediction accuracy compared to AM-FNO. We hypothesize that AFNO's lower accuracy is due to using the same MLP to evolve all frequency modes of the input functions, potentially limiting expressiveness, whereas AM-FNO transforms different frequency modes separately. Further discussion on AFNO will be included in the revised version.
>
> |Benchmark|AM-FNO(MLP)|AM-FNO(KAN)|AFNO|
> |------|--------|---------|--------|
> |Darcy|4.21e-3|4.28e-3|3.17e-2|
> |Airfoil|5.64e-3|6.06e-3|9.88e-3|
>
> Q1: The smoothness about MLP.
>
> The term "more smoothly" in our context indicates that our method achieves smoother evolution compared to FNO, which parameterizes the kernel value point-by-point. This pointwise parameterization may ignore correlations between frequency modes, leading to a less smooth kernel. In contrast, our method uses an MLP to approximate the mapping between frequency modes and their corresponding kernel values, resulting in a smoother representation. As mentioned in W1, directly using an MLP might struggle to capture the complex structure of the kernel, but the orthogonal embedding helps address this issue.  However, we believe the improved performance is due to the increased expressiveness to model kernel functions, rather than just a matter of smoothness.
>
> Q2: The central symmetry in Fourier space
>
> Yes. We ensure this central symmetry by obtaining the value at (-k_x, -k_y) from the NN prediction at (k_x, k_y) directly.
>
> Q3:  GPU memory and training time comparison.
>
> We provide a comparison of the training time and memory consumption for AM-FNO and FNO (with all modes retained) in the table below, tested on the 2D Darcy benchmark with a resolution of $421 \times 421$. The results indicate that AM-FNO (MLP) exhibits both reduced memory usage and shorter training times compared to FNO, which is attributable to its lower complexity. Although AM-FNO (KAN) demonstrates increased training time due to its architectural design, it still benefits from lower memory consumption. We conjecture that this advantage is particularly evident when solving PDEs with high resolution and high dimensions. A comprehensive analysis will be included in the updated version of the paper.
>
> |Model|Memory|Train Time|
> |-----|------|------|
> |AM-FNO(MLP)|**9.7G**|**43.1s**|
> |AM-FNO(KAN)|13.5G|83.8s|
> |FNO|14.9G|45.6s|
>
> We are sincerely grateful for your valuable insights, which we firmly believe will significantly enhance the quality of our manuscript. If you have any more questions or find some mistakes, please feel free to correct us.
>
> [1] S Qian, YC Lee, RD Jones, CW Barnes, and K Lee. Function approximation with an orthogonal basis net. In 1990 IJCNN International Joint Conference on Neural Networks, pages 605–619. IEEE, 1990.

---

> > ### Comment · Reviewer_DLAH · 2024-08-09
> >
> > Thank you for your response.
> >
> > For the **motivation (W1)**, I can see your thoughts on why MLP without the orthogonal basis is not good enough. However, the current motivation remains ambiguous. The statement that 'MLPs struggle to capture the complex, non-linear nature of the kernel function' appears speculative. It would be more convincing if you could provide empirical evidence or a formal argument to support this statement.
> >
> > This is why I mentioned the **spectral bias (Q1) of MLPs**, a well-known property for MLPs with both empirical and theoretical evidence. Following this concept, a function's complexity and non-linearity are linked to its high-frequency components: the more complex and non-linear the function, the more high-frequency components it has. Then, using the **Fourier basis (W2)** as embedded functions to improve the high-frequency property of MLPs can be well supported by the literature.
> >
> > I asked about **smoothness (Q1)** because it describes a function's spectral properties—smoother functions have fewer high-frequency components. In your paper, you aimed to make the kernel function smoother (with fewer high-frequency components) but then added an orthogonal basis to the MLP to represent more complex, non-linear functions (with more high-frequency components). This seems contradictory to me.
> >
> > I'm curious why you haven't discussed the spectral bias of MLPs in your paper, given its significance. As I mentioned, previous research on spectral bias and the Fourier basis provides a strong explanation for why an orthogonal basis helps MLPs represent more complex functions.
> >
> > For **AFNO (W3)**, it would be beneficial to present its results across all datasets used in this paper (Table 2). Additionally, a more thorough comparison of AM-FNO and AFNO would be helpful, such as including ablation studies or an efficiency analysis of both methods. If AFNO is just another unrelated neural operator, it's fine to simply guess why it performs worse than AM-FNO. However, given the similar motivation and implementation between AFNO and AM-FNO, a deeper analysis is necessary.
> >
> > Regarding **GPU memory and training time (Q3)**, could you please provide more detailed hyperparameters for the models? I’m having trouble understanding why AM-FNO requires less GPU memory and training time compared to FNO. As I understand it, AM-FNO uses MLP to reduce the number of parameters, but this should increase computation. Whether you use shared or different weights for different modes, each Fourier layer still needs to map k modes from input to output. Given that FNO truncates high-frequency modes, it seems FNO should require less computation than AM-FNO.

---

> > > ### Author Response · Authors · 2024-08-10
> > > **Reply to Reviewer DLAH**
> > >
> > > Thanks for your detailed feedback.
> > >
> > > # Motivation
> > >
> > > We hadn't initially considered spectral bias a significant reason for explaining our work. Thus we greatly appreciate your suggestion regarding spectral bias and concur that it provides a well-founded perspective for interpreting our method. We will expand the discussion on it in the revised version.
> > >
> > > # AFNO
> > >
> > > Due to the time constraints of the rebuttal phase, we have only presented results for AFNO on two benchmarks. In the revised version, we will provide a more comprehensive discussion along with additional empirical results pertaining to AFNO.
> > >
> > > # GPU memory and training time
> > >
> > > The models are all with 4 layers and 32 widths. As mentioned above, FNO retains all frequency modes ($421 \times 211$ due to central symmetry), while AM-FNO (MLP) utilizes 32 orthogonal basis functions, and AM-FNO (KAN) employs 32 local basis functions.
> > >
> > > Regarding GPU memory usage, FNO requires significantly more memory to store model weights, gradients, and the optimizer, owing to its large parameter count. Although AM-FNOs necessitate memory to store the hidden state used to derive the kernel, the overall memory requirement remains lower.
> > >
> > > In terms of training time, AM-FNOs indeed demand more computation compared to FNO. However, due to the large kernel size in FNO, a substantial amount of memory reads, writes, and gradient computations are required for parameter updates. Consequently, FNO demands more training time, when training with high-resolution and high-dimensional data.
> > >
> > > We are grateful for your valuable feedback, which we believe will significantly enhance the quality of our paper. Should you have any further questions, please feel free to reach out.

---

> ### Comment · Reviewer_DLAH · 2024-08-10
>
> Thank you for your response. I understand that the rebuttal period is only one week, and it's not feasible to run many experiments or make significant revisions to the paper. Given the limited time for discussion, I'll focus on my major concern: **GPU memory and training time**.
>
> Using the full frequency modes (421) for FNO in your comparison of GPU memory and training time is problematic. Previous studies often use a truncation mode of around 12 for FNO, for three main reasons: (1) Keeping too many modes increases computational costs considerably. (2) For PDE data such as Darcy flow, with minimal high-frequency components, using so many modes (421) is redundant. (3) Using too many modes can actually damage FNO's performance.
>
> Therefore, using FNO with a truncation mode of 421 is a very weak baseline in terms of accuracy, GPU memory usage, and time consumption. The current comparison appears to increase FNO's parameters to an unreasonably large number, then claims that AM-FNO uses fewer parameters and GPU memory than this excessively large FNO.
>
> Compared to the commonly used FNO, AM-FNO's architecture is expected to demand more GPU memory and longer training time, which could significantly limit its application for high-dimensional PDE data. Additionally, AM-FNO cannot guarantee it has fewer parameters than the commonly used FNO.
>
> I recommend that the authors provide a more detailed discussion on the efficiency of AM-FNO to avoid presenting potentially misleading results.

---

> > ### Author Response · Authors · 2024-08-11
> > **Reply to Reviewer DLAH**
> >
> > Thanks for your prompt response.
> >
> > We understand your concern regarding GPU memory usage and training time. Below, we provide a comparison of GPU memory and training time between our models and baselines, including FNO with 12 modes, tested on the same benchmark. While our models require more training time and memory than the standard FNO due to the use of MLPs or KANs, the difference is relatively minor. We appreciate your suggestion and will include a more detailed discussion on the efficiency of our method.
> >
> > | Model| AM-FNO (MLP)| AM-FNO (KAN) | FNO | U-FNO | OFormer | LSM | F-FNO |
> > |------|------|------|------|------|------|------|------|
> > |Train Time (s/epoch) |  1.2  | 2.1  | 0.9   |  1.6  | 16.5 |   2.1 |  1.1  |
> > |Memory (M) | 1850 |  2066  |  1212  | 1444 | 16090   |  1894  | 1126  |
> > |Params. (M) |  1.1  |  1.5  |  2.4  |  2.6  |  1.3  |  4.8  |  0.2 |
> >
> > If you have any further questions, we are pleased to discuss them.

---

> > > ### Comment · Reviewer_DLAH · 2024-08-11
> > >
> > > Thank you for providing these results. However, they don’t seem to align with the table you previously shared in the rebuttal or with Table 6 in your paper. Could you confirm which dataset is used for this test and what the data resolution is?

---

> > > > ### Author Response · Authors · 2024-08-11
> > > > **Reply to Reviewer DLAH**
> > > >
> > > > The results were tested on the Darcy benchmark (with an $85 \times 85$ resolution) using a batch size of 16. The hyperparameters were consistent with those used in the main experiments presented in our paper (Table 2).
> > > >
> > > > If you have any further questions, we are pleased to discuss them.

---

> > > > > ### Comment · Reviewer_DLAH · 2024-08-11
> > > > >
> > > > > So, is it correct that in each Fourier layer, AM-FNO processes 85 * 43 = 3,655 modes while FNO processes 23 * 12 = 276 modes? And yet, AM-FNO only takes about 1.33 times the training time and 1.5 times the memory to handle approximately 13 (3,655 / 276) times the modes? If that's true, I'd say AM-FNO's efficiency is impressive.
> > > > >
> > > > > Could you also comment on why AM-FNO outperforms FNO? Is it mainly because (1) AM-FNO doesn't truncate high-frequency modes or (2) AM-FNO uses MLP for all modes? I believe you could gain insights by running a test with AM-FNO using the same truncation mode as FNO. If time doesn’t allow for this test, any comments based on your understanding would also be helpful.

---

> ### Author Response · Authors · 2024-08-11
> **Reply to Reviewer DLAH**
>
> We clarify that AM-FNOs use 85×43 modes, whereas FNO uses 2×12×12 modes. AM-FNO requires MLPs to generate the kernel and perform matrix-vector multiplication with a larger kernel. However, both operations can be parallelized for high efficiency.
>
> We present the performance of AM-FNOs **trained with $24 \times 12$ modes** (requested by Reviewer Z566) on Darcy benchmark, as shown in the table below. Even with truncated modes, AM-FNOs still significantly outperform FNO, demonstrating the effectiveness of MLP parameterization. While maintaining all modes can further improve performance, the impact is relatively modest on this benchmark.
>
> | Modes  | AM-FNO(MLP)  | AM-FNO(KAN) |   FNO   |
> |----------| ---------|---------| ------- |
> | 24*12 | 4.72e-3 | 4.78e-3 | 1.08e-2 |
> | 85*43 | 4.21e-3 | 4.28e-3 | - |
>
> If you have any further questions, we are pleased to discuss them.

---

> > ### Comment · Reviewer_DLAH · 2024-08-11
> >
> > I see, and I'm now convinced that AM-FNO's design is valuable in terms of both accuracy and efficiency.
> >
> > However, the authors should consider revising the abstract and introduction. Based on the available experiments, the major benefit of AM-FNO appears to come from the MLP parameterization, not from using full frequency modes. Yet, the current motivation in the abstract and introduction focuses primarily on the issue of high-frequency truncation in FNO. Since there's evidence that increasing the number of modes in FNO may actually damage its performance, using full frequency modes doesn't provide a strong enough motivation. I suggest focusing the motivation on the curse of dimensionality, highlighting AM-FNO is designed to reduce the number of parameters in FNO. This motivation holds whether truncation is used or not and is more consistent with the experimental results.
> >
> > I also recommend adding a discussion and ablation study on AM-FNO's truncation frequency across other datasets in this paper, similar to the test on Darcy flow. This is crucial for helping readers understand why AM-FNO performs better. The authors should clarify that AM-FNO doesn't necessarily require full frequency modes, as experiments demonstrate that AM-FNO with high-frequency truncation is actually a very competitive method in both accuracy and efficiency.
> >
> > I appreciate the authors' efforts to address my concerns. I would be happy to raise my score if the authors could revise the paper based on our discussion.

---

> > > ### Author Response · Authors · 2024-08-11
> > > **Reply to Reviewer DLAH**
> > >
> > > We appreciate your valuable suggestions regarding the motivation for our study. We will revise the abstract and introduction to incorporate your feedback. We will also include empirical results and a discussion on frequency truncation in the revised version.
> > >
> > > Thank you sincerely for the constructive discussion, which will help enhance our work. We will carefully consider the points mentioned and make revisions accordingly.

---

> > > > ### Comment · Reviewer_DLAH · 2024-08-11
> > > >
> > > > I've raised my score to 6 since the authors have addressed all my major concerns.

---

> > > > > ### Author Response · Authors · 2024-08-12
> > > > > **Reply to Reviewer DLAH**
> > > > >
> > > > > Thank you very much for the valuable feedback and the improved score. We will thoroughly consider your suggestions and revise our paper accordingly.

---

### Author Rebuttal · Authors · 2024-08-07

We would like to express our gratitude for the thoughtful reviews. We are pleased that the reviewers found our paper to be **overall well-written, with clear descriptions of the methods** (Reviewer DLAH), that our method is **original and intuitive** (Reviewer DLAH), **novel** (Reviewer okt7), our theoretical analysis is **solid** (Reviewer okt7), and that our experimental results **consistently outperform various baselines/achieve better performance** (Reviewer DLAH, Reviewer NuVs).

1) To address the concerns of Reviewers DLAH, NuVs, and okt7, we provide a comparison of GPU memory usage and inference/training times between our method and FNO, tested on the 2D Darcy benchmark with a resolution of 421 $\times$ 421.

2) We report the repeated results on NS-2D and CFD-2D benchmarks to address the concern of Reviewer NuVs and kDgT.

3) We add more baselines including AFNO, GNOT and ONO, and more benchmarks (3D Plasticity and Elasticity in point cloud [1]) to address the concern of Reviewer DLAH, Z566, okt7.

**Table A: Comparison of GPU Memory Usage and Inference/Training Times.**

|Model|Memory|Train Time|Inf Time|
|-----|------|------|------|
|AM-FNO(MLP)|**9.7G**|**43.1s**|2.4s|
|AM-FNO(KAN)|13.5G|83.8s|2.2s|
|FNO|14.9G|45.6s|2.2s|

**Table B: Comparison on Darcy and Airfoil benchmarks with additional baselines.**

|Benchmark|AM-FNO(MLP)|AM-FNO(KAN)|GNOT|ONO|AFNO|
|------|--------|---------|--------|--------|--------|
|Darcy|**4.21e-3**|4.28e-3|1.05e-2|7.20e-3|3.17e-2|
|Airfoil|**5.64e-3**|6.06e-3|7.57e-3|5.60e-3|9.88e-3|

**Table C: Comparison on Elasticity benchmark.**

|Benchmark|AM-FNO(MLP)|AM-FNO(KAN)|Geo-FNO|LSM|GNOT|ONO|
|------|--------|---------|--------|--------|--------|--------|
|Elasticity|2.03e-2|2.10e-2|2.29e-2|2.25e-2|8.6e-3|1.18e-2|

**Table D: Repeated results on NS2d and CFD2d benchmarks.**

|Benchmark|AM-FNO(MLP)|AM-FNO(KAN)|
|------|--------|---------|
|NS-2D|8.53e-2 ± 7.48e-4|1.04e-1± 3.27e-3|
|CFD-2D|2.21e-3 ± 4.13e-5|2.75e-3 ± 8.96e-5|

**Reference**

[1] Li, Z., Huang, D. Z., Liu, B., & Anandkumar, A. (2023). Fourier neural operator with learned deformations for pdes on general geometries. Journal of Machine Learning Research, 24(388), 1-26.

[2] Li, Z., Zheng, H., Kovachki, N., Jin, D., Chen, H., Liu, B., ... & Anandkumar, A. (2024). Physics-informed neural operator for learning partial differential equations. ACM/JMS Journal of Data Science, 1(3), 1-27.

[3] Guibas, J., Mardani, M., Li, Z., Tao, A., Anandkumar, A., & Catanzaro, B. (2021). Adaptive fourier neural operators: Efficient token mixers for transformers. arXiv preprint arXiv:2111.13587.

[4] Yu, R., Yu, W., & Wang, X. (2024). Kan or mlp: A fairer comparison. arXiv preprint arXiv:2407.16674.

---

### Author Response · Authors · 2024-08-14
**Rebuttal Summary**

As the rebuttal period ends, we would like to briefly summarize some key points from the authors' perspective.

Firstly, we would like to express our gratitude for the thoughtful reviews and discussions. We are pleased that the reviewers found our paper to be **overall well-written**, **with clear descriptions of the methods** (Reviewer DLAH), that our method is **original and intuitive** (Reviewer DLAH) and **novel** (Reviewer okt7), our theoretical analysis is **solid** (Reviewer okt7), and that our experimental results **consistently outperform various baselines/achieve better performance** (Reviewer DLAH, Reviewer NuVs).

Secondly, we would like to conclude all the efforts we made during the rebuttal phase.

- Experiments with additional baselines: Include experiments with additional baselines, particularly AFNO.

- Experiments on additional benchmarks: Expand to include higher-dimensional settings and unstructured meshes.

- Efficiency comparison: Provide a comprehensive analysis including training time, inference time, and GPU memory usage.

- Clarify the differences between SNO and our method.

- Clarify our motivation about the orthogonal embedding.

- Experiment with our model using truncated modes or one layer.

- Explain why we did not report the repeated results.

- Explain why we did not use a validation dataset.


Based on the discussions so far, we have addressed most concerns raised by reviewers.
Specifically, Reviewer DLAH has acknowledged that we have addressed all the major concerns and is convinced that our design "is valuable in terms of both accuracy and efficiency".
Reviewer Z566 also recognized that the concerns were addressed and agreed that our method improves performance.

To further improve our work and address the reviewer's concerns, we would make the following revisions:

- We will revise our motivation and incorporate spectral bias to explain the orthogonal embedding.  (Reviewer DLAH & Z566 & NuVs)

- We will include additional baselines (AFNO, DeepONet, SNO, etc.) and introduce more benchmarks involving higher dimensions and unstructured meshes. (Reviewer DLAH & Z566 & okt7)

- We will provide a more rigorous version of the theorem concerning orthogonal basis functions. (Reviewer kDgT)

- We will provide a more comprehensive analysis of efficiency and frequency truncation and discuss its limitations. (Reviewer DLAH & Z566 & NuVs & okt7)

We want to express many thanks to reviewers for asking questions or even challenging us. We will carefully consider all suggestions. This engagement in the rebuttal phase has been invaluable for improving our paper and deepening our understanding of our work.

---

### Decision · Program_Chairs · 2024-09-25

**Decision:**

Accept (poster)

**Comment:**

The paper introduced Amortized Fourier Neural Operators (AM-FNO), which aim to alleviate the problem of limited frequency modes in FNOs. The key contribution of the work is to introduce a neural network-based function to map frequency to the Fourier transform of the kernel function. This is not as straightforward, and the authors carefully map these values using two neural networks (identical, but these map to real and imaginary parts respectively) and also augment the NN with orthogonal embedding functions. This parametrization leads to improvements both in terms of accuracy and efficiency. Therefore, we recommend an acceptance.

We recommend that the authors to incorporate the edits suggested by the reviewers including the following.

1. Consider defining "." in eq (3)
2. line 111 -- "On the other hand, this guarantees the kernel function to evolve more smoothly as the frequency mode changes, due to the smoothness of NN". Consider clarifying how smoothness is ensured since smoothness is measured by differentiability and NN in general are non-smooth. Next, "more smoothly" does not convey technical meaning, since functions are either smooth or they are not (a binary). Do you intend to use the continuity of the function?
3. Consider clarifying what kind of activations are used in the experiments.
4. Line 135 -- Consider adding support for or clarify the statement "vanilla MLPs lack effective inductive bias for function approximation." Since this is a contradiction to universal approximation property (stated in the previous statement and also , Hornik et al. (1989), Cybenko (1989), Funahashi (1989). Also, clarify or provide support for "vanilla MLPs".
5. Consider adding a line about the reason for calling the method "Amortized".
6. line 145 and 137 -- "we can implement the NN in defining Rp,q as a linear transformation upon a set of orthogonal functions" -- Since the discussion here is about MLPs this seems like a major switch in context since the entire discussion is centered around NNs. Moreover, Fourier basis can serve as orthogonal basis. Consider clarifying this. Also, consider making it clear (via mathematical expressions) what "augment" in "we propose to augment the MLP with orthogonal embedding functions to construct AM-FNO.", and including a clear rationale for employing these (some of the justification included in the rebuttal phase were not very direct and clear).
7. A description of the hyper-parameters and baseline settings should be included in the manuscript for reproducibility.
8. Consider correcting the discussion about the computational complexity in the manuscript based on the discussion with the reviewers in the rebuttal phase.
9. Any new data and tables which were presented during the rebuttals should be added to the manuscript/supplementary with the corresponding discussion.


Hornik et al. (1989). Multilayer feedforward networks are universal approximators. Neural networks, 2(5), 359-366.
Cybenko (1989). Approximation by superpositions of a sigmoidal function. Mathematics of control, signals and
systems, 2(4):303–314, 1989.
Funahashi (1989) Funahashi, K.-I. On the approximate realization of continuous mappings by neural networks. Neural networks, 2(3):183–192, 1989.